# The normalization model predicts responses in the human visual cortex during object-based attention

Narges Doostani[1], Gholam-Ali Hossein-Zadeh[1,2], Maryam Vaziri-Pashkam[3]*

[1]School of Cognitive Sciences, Institute for Research in Fundamental Sciences, Tehran, Iran; [2]School of Electrical Engineering, University of Tehran, Tehran, Iran; [3]Laboratory of Brain and Cognition, National Institute of Mental Health, Bethesda, United States

*For correspondence:
maryam.vaziri-pashkam@nih.gov

Competing interest: The authors declare that no competing interests exist.

**Abstract** Divisive normalization of the neural responses by the activity of the neighboring neurons has been proposed as a fundamental operation in the nervous system based on its success in predicting neural responses recorded in primate electrophysiology studies. Nevertheless, experimental evidence for the existence of this operation in the human brain is still scant. Here, using functional MRI, we examined the role of normalization across the visual hierarchy in the human visual cortex. Using stimuli form the two categories of human bodies and houses, we presented objects in isolation or in clutter and asked participants to attend or ignore the stimuli. Focusing on the primary visual area V1, the object-selective regions LO and pFs, the body-selective region EBA, and the scene-selective region PPA, we first modeled single-voxel responses using a weighted sum, a weighted average, and a normalization model and demonstrated that although the weighted sum and weighted average models also made acceptable predictions in some conditions, the response to multiple stimuli could generally be better described by a model that takes normalization into account. We then determined the observed effects of attention on cortical responses and demonstrated that these effects were predicted by the normalization model, but not by the weighted sum or the weighted average models. Our results thus provide evidence that the normalization model can predict responses to objects across shifts of visual attention, suggesting the role of normalization as a fundamental operation in the human brain.

## Editor's evaluation

This study on object-based attention furthers of understanding of the role of normalization across the visual hierarchy in the human visual cortex. The authors provide solid functional MRI evidence that supports their claims, demonstrating that the normalization model predicts the observed effect when participants selectively attend to one of two stimulus categories. The paper is an important contribution to the fields of perceptual and cognitive neuroscience.

## Introduction

The brain makes use of fundamental operations to perform neural computations in various modalities and different regions. Divisive normalization has been proposed as one of these fundamental operations. Under this computation, the response of a neuron is determined based on its excitatory input divided by a factor representing the activity of a pool of nearby neurons (*Heeger, 1992*; *Carandini et al., 1997*; *Carandini and Heeger, 2011*). Normalization was first introduced based on responses in the cat primary visual cortex (*Heeger, 1992*), and evidence of its operation in higher regions of the

monkey visual cortex has also been demonstrated both during passive viewing (*Bao and Tsao, 2018*) and when attention is directed towards a stimulus (*Reynolds and Heeger, 2009*; *Lee and Maunsell, 2010*; *Ni et al., 2012*; *Ni and Maunsell, 2019*).

Normalization has also been proposed as a critical operation in the human brain based on evidence demonstrating the sublinear addition of responses to multiple stimuli in the visual cortex (*Bloem and Ling, 2019*). Nevertheless, in lieu of directly testing the normalization model to resolve multiple-stimulus representation, several previous studies have shown that a weighted average model can account for multiple-stimulus responses in the monkey brain (*Zoccolan et al., 2005*; *Macevoy and Epstein, 2009*; *Reddy et al., 2009*; *Kliger and Yovel, 2020*). The only exception is a recent electro-physiology study, which showed that in the category-selective regions of the monkey brain, a winner-take-all, but not averaging, rule can explain neural responses in many cases (*Bao and Tsao, 2018*). *Bao and Tsao, 2018* further demonstrated that the normalization model predicts such winner-take-all behavior. It is not clear whether this discrepancy has emerged as a result of different explored regions of the brain, or due to the diversity in stimuli or the task performed by the participants.

In addition to regional computations for multiple-stimulus representation, the visual cortex relies on top-down mechanisms such as attention to select the most relevant stimulus for detailed processing (*Moran and Desimone, 1985*; *Desimone and Duncan, 1995*; *Chun et al., 2011*; *Baluch and Itti, 2011*; *Noudoost et al., 2010*; *Maunsell, 2015*; *Thiele and Bellgrove, 2018*; *Itthipuripat et al., 2014*; *Moore and Zirnsak, 2017*; *Buschman and Kastner, 2015*). Attention works through increasing the response gain (*Treue and Martínez Trujillo, 1999*; *McAdams and Maunsell, 1999*) or contrast gain (*Reynolds et al., 2000*; *Martínez-Trujillo and Treue, 2002*) of the attended stimulus. Previous studies have demonstrated how the normalization computation accounts for these observed effects of attention in the monkey brain. They have suggested that normalization attenuates the neural response in proportion to the activity of the neighboring neuronal pool (*Reynolds and Heeger, 2009*; *Ni et al., 2012*; *Boynton, 2009*; *Lee et al., 2009*). These studies have focused on space-based (*Reynolds and Heeger, 2009*; *Ni et al., 2012*; *Lee et al., 2009*) or feature-based (*Ni and Maunsell, 2019*) attention. While it has been suggested that these different forms of attention affect neural responses in similar ways, there exist distinctions in their reported effects, such as different time courses (*Hayden and Gallant, 2005*) and the extent to which they affect different locations in the visual field (*Serences and Boynton, 2007*; *Womelsdorf et al., 2006*), suggesting that there are common sources as well as differences in modulation mechanisms between these forms of attention (*Ni and Maunsell, 2019*). This leaves open the question of whether normalization can explain the effects of object-based attention.

In the human visual cortex, normalization has been speculated to underlie response modulations in the presence of attention, with evidence provided both by behavioral studies of space-based (*Herrmann et al., 2010*) and feature-based (*Herrmann et al., 2012*; *Schwedhelm et al., 2016*) attention, as well as neuroimaging studies of feature-based attention (*Bloem and Ling, 2019*). Although previous studies have qualitatively suggested the role of normalization in the human visual cortex (*Bloem and Ling, 2019*; *Kliger and Yovel, 2020*; *Itthipuripat et al., 2014*; *Zhang et al., 2016*), evidence for directly testing the validity of the normalization model in predicting human cortical responses in a quantitative way remains scarce. A few studies have demonstrated the quantitative advantage of normalization-based models compared to linear models in predicting human fMRI responses using gratings, noise patterns, and single objects (*Kay et al., 2013a*; *Kay et al., 2013b*), as well as moving checkerboards (*Aqil et al., 2021*; *Foster and Ling, 2021*). However, whether normalization can also be used to predict cortical responses to multiple objects, and if and to what extent it can explain the modulations in response caused by attention to objects in the human brain remain unanswered.

To fill this gap and to explore the discrepancies reported about multiple-stimulus responses, here, we aimed to evaluate the predictions of the normalization model against observed responses to visual objects in several regions of the human brain in the presence and absence of attention. In an fMRI experiment using conditions with isolated and cluttered stimuli and recording the response with or without attention, we provide a comprehensive account of normalization in different regions of the visual cortex, showing its success in adjusting the gain related to each stimulus when it is attended or ignored. We also demonstrate that normalization is closer to average in the absence of attention, as previously reported by several studies (*Zoccolan et al., 2005*; *Macevoy and Epstein, 2009*; *Kliger and Yovel, 2020*), but that the results of the weighted average model and the normalization model diverge to a greater extent in the presence of attention. Our work in the human brain, along with

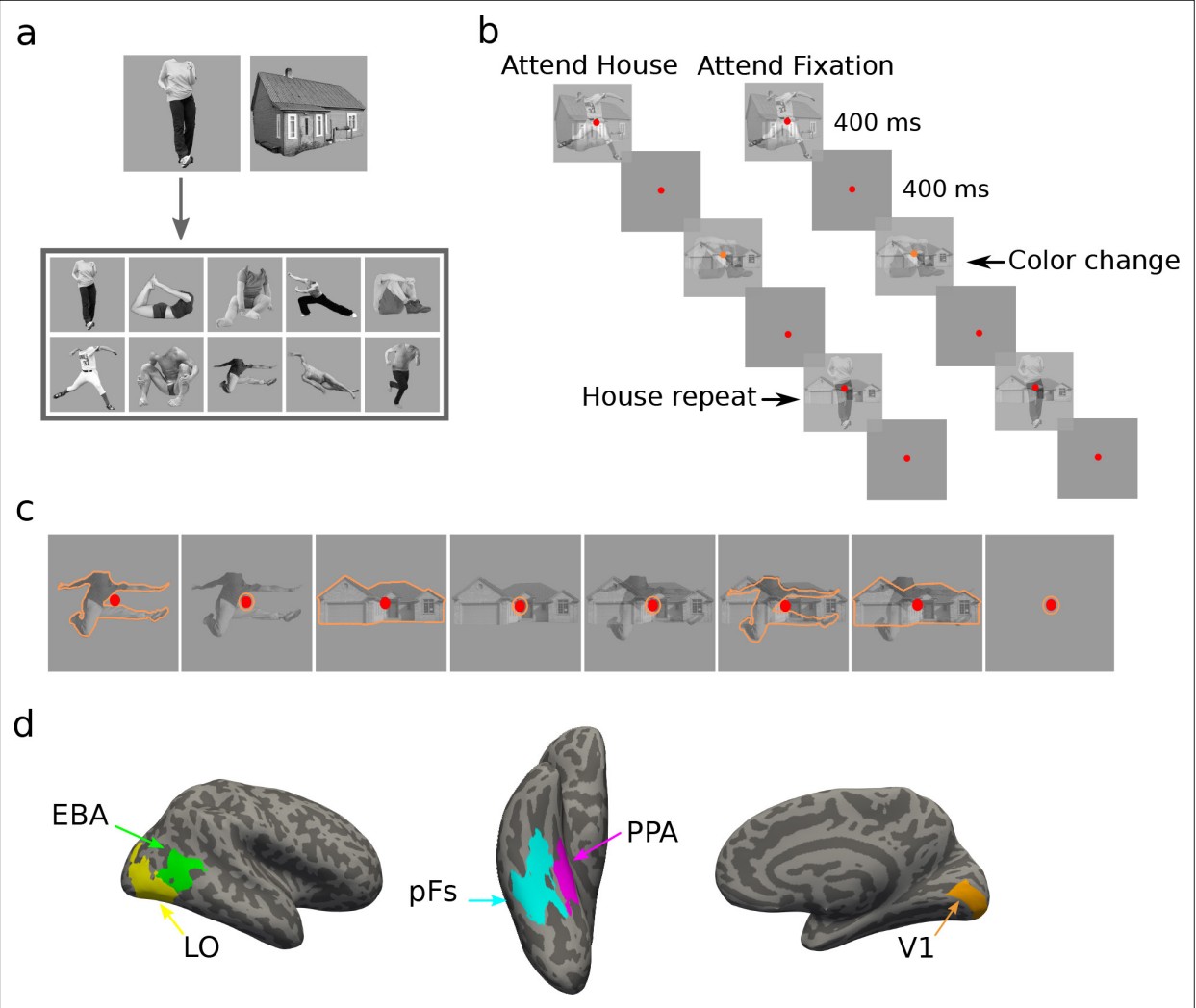

**Figure 1.** Stimuli, paradigm, and regions of interest. (**a**) The two stimulus categories (body and house), with the ten exemplars of the body category. (**b**) Experimental paradigm including the timing of the trials and the inter-stimulus interval. In the example block depicted on the left, both stimulus categories were presented, and the participant was cued to attend to the house category. The two stimuli were superimposed in each trial, and the participant had to respond when the same stimulus from the house category appeared in two successive trials. The color of the fixation point randomly changed in some trials from red to orange, but the participants were asked to ignore the color change. The example block depicted on the right illustrates the condition in which stimuli were ignored and participants were asked to attend to the fixation point color, and respond when they detected a color change. Subjects were highly accurate in performing these tasks (see *Figure 1—figure supplement 1*). (**c**) The eight task conditions in each experimental run. For illustration purposes, we have shown the attended category in each block with orange outlines. The outlines were not present in the actual experiment. (**d**) Regions of interest for an example participant, including the primary visual cortex V1, the object-selective regions LO and pFs, the body-selective region EBA, and the scene-selective region PPA.

The online version of this article includes the following figure supplement(s) for figure 1:

**Figure supplement 1.** Behavioral performance of participants for the main experiment.

previous studies of normalization in the monkey and human brain, suggests the role of normalization as a canonical computation in the primate brain.

## Results

### Attention modulates responses to isolated and paired stimuli

In a blocked-design fMRI paradigm, human participants (N = 19) viewed semi-transparent gray-scale stimuli from the two categories of houses and human bodies (*Figure 1a*). Each experimental run consisted of one-stimulus (isolated) and two-stimulus (paired) blocks, with attention directed either

to an object stimulus or to the color of the fixation point. There was an additional fixation color block in each run with no object stimuli, in which the participants were asked to attend to the fixation point color. The experiment, therefore, had a total number of eight conditions (four isolated, three paired, and one fixation conditions, see *Figure 1c*). In paired blocks, we superimposed the two stimuli to minimize the effect of spatial attention and force participants to use object-based attention (*Figure 1b and c*). Participants were asked to perform a one-back repetition detection task on the attended object, or a color change detection task on the fixation point (*Figure 1b*, see Methods for details). Independent localizer runs were used to localize the primary visual cortex (V1), the object-selective regions in the lateral occipital cortex (LO) and posterior fusiform gyrus (pFs), the extrastriate body area (EBA), and the parahippocampal place area (PPA) for each participant (*Figure 1d*).

Each task condition was named based on the presented stimuli and the target of attention, with B and H denoting the presence of body and house stimuli, respectively, and the superscript *at* denoting the target of attention. Therefore, the seven task conditions include $B^{at}$, $B^{at}H$, $BH^{at}$, $H^{at}$, B, H, and BH. For instance, the $H^{at}$ condition refers to the isolated house condition with attention directed to house stimuli, and the BH condition refers to the paired condition with attention directed to the fixation point color. Overall, the average accuracy was higher than 86% in all conditions. Averaged across participants, accuracy was 94%, 89%, 86%, 93%, 94%, 96%, 95%, and 96% for $B^{at}$, $B^{at}H$, $BH^{at}$, $H^{at}$, B, H, and BH conditions and the fixation block with no stimulus, respectively. A one-way ANOVA test across conditions showed a significant effect of condition on

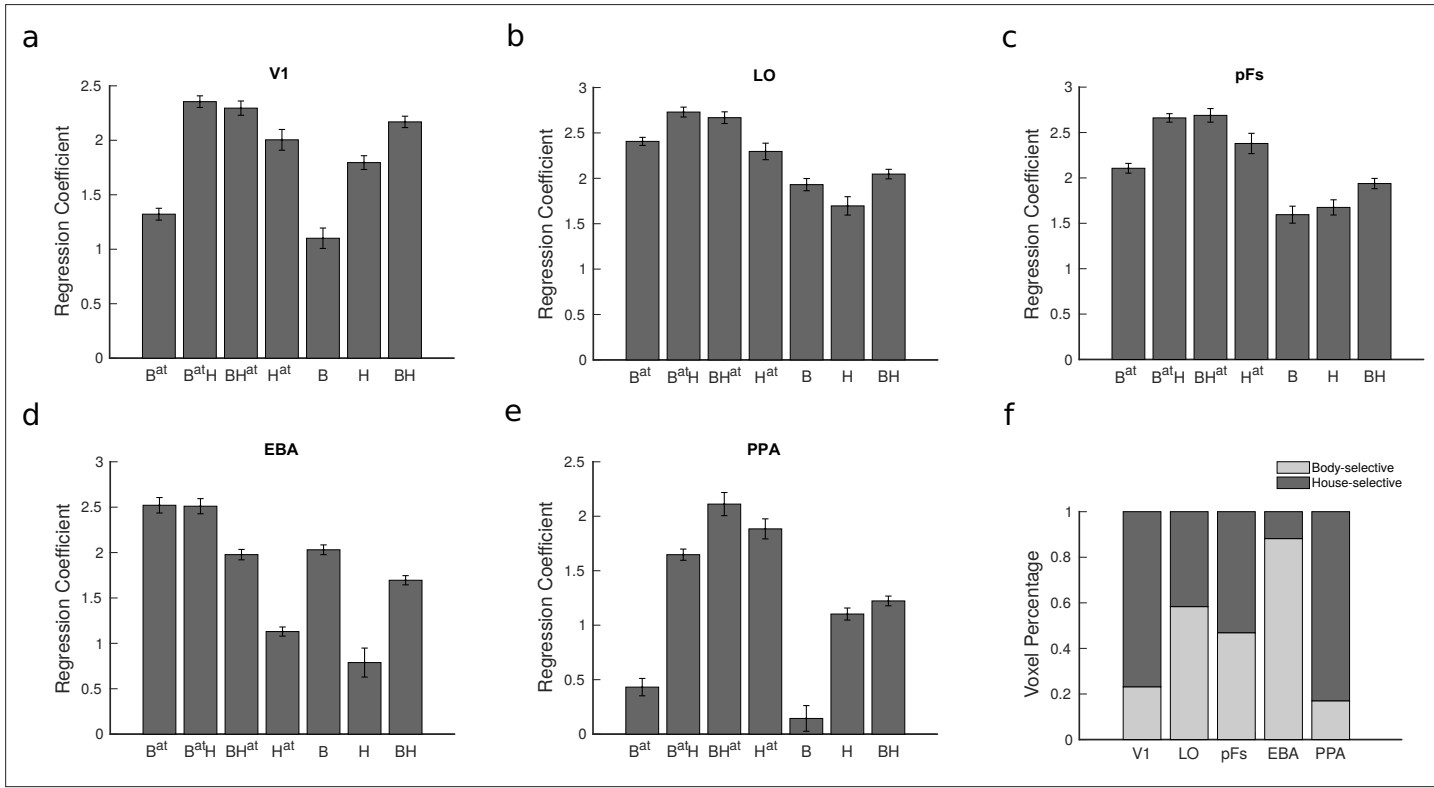

**Figure 2.** Average fMRI regression coefficients and voxel preference for the two categories in all regions of interest (ROIs). (**a–e**) Average fMRI regression coefficients for each condition are illustrated in the five ROIs. Each condition's label denotes the presented stimuli and the target of attention, with B and H, respectively, denoting the presence of body and house stimuli and the superscript *at* denoting the target of attention. Therefore, the seven task conditions include $B^{at}$, $B^{at}H$, $BH^{at}$, $H^{at}$, B, H, and BH. For instance, the $H^{at}$ condition refers to the isolated house condition with attention directed to houses, and the BH condition refers to the paired condition with attention directed to the fixation point color. Error bars represent standard errors of the mean for each condition, calculated across participants after removing the overall between-subject variance. N = 19 human participants. (**f**) The ratio of voxels preferring bodies and houses in each ROI. Both the regression coefficients and the voxel preference ratios were consistent across odd and even runs (see *Figure 2—figure supplement 1* and *Figure 2—figure supplement 1*).

The online version of this article includes the following figure supplement(s) for figure 2:

**Figure supplement 1.** Average fMRI regression coefficients for each condition and voxel preference for the two categories in all ROIs, illustrated seperately for odd and even runs.

accuracy ($F(7, 126) = 8.24$, $p < 0.0001$) and reaction time ($F(7, 126) = 22.57$, $p < 0.0001$). As expected, post-hoc t-tests showed that this was due to lower performance in the B$^{at}$H and BH$^{at}$ conditions (see *Figure 1—figure supplement 1*). There was no significant difference in performance between any other conditions ($ps > 0.07$, *corrected*).

To examine the cortical response in different task conditions, we fit a general linear model and estimated the regression coefficients for each voxel in each condition. *Figure 2* illustrates the average voxel coefficients for different conditions in the five regions of interest (ROIs), including V1, LO, pFs, EBA, and PPA. Note that we have not included the responses related to the fixation block with no stimulus since this condition was only used to select the voxels that were responsive to the presented stimuli in each ROI (see Methods). We observed that the average voxel coefficients related to the four conditions in which attention was directed to the body or the house stimuli (the first four conditions, B$^{at}$, B$^{at}$H, BH$^{at}$, H$^{at}$) were generally higher than the response related to the last three conditions (B, H, and BH conditions) in which the body and house stimuli were unattended ($ts > 4$, $ps < 0.01$, *corrected*). This is in agreement with previous research indicating that attention to objects increases their cortical responses (*Reddy et al., 2009*; *Roelfsema et al., 1998*; *O'Craven et al., 1999*).

Looking more closely at the results in the regions EBA and PPA that have strong preferences for body and house stimuli, respectively, it seems that the effect of attention interacts with the regions' preference. For instance, in the body-selective region EBA, the response to attended body stimuli in isolation is similar to the response to attended body stimuli paired with unattended house stimuli (compare B$^{at}$ and B$^{at}$H bars). On the other hand, the response to attended house stimuli in the isolated condition is significantly less than the response to attended house stimuli paired with unattended body stimuli. We can observe similar results in PPA, but not in V1 or the object-selective regions LO and pFs. But note that the latter three regions do not have strong a preference for one stimulus versus the other. Therefore, in order to examine the interaction between attention and preference more closely, we determined preferences at the voxel level in all ROIs.

We defined the preferred (P) and null (N) stimulus categories for each voxel in each ROI according to the voxel's response to isolated body and isolated house conditions. *Figure 2f* shows the percentage of voxels in each region that were selective to bodies and houses averaged across participants. As illustrated in the figure, in the object-selective regions LO and pFs, almost half of the voxels were selective to each category, while in the EBA and PPA regions, the general preference of the region prevailed (Even though these regions were selected based on their preference, the noise in the fMRI data and other variations due to imperfect registration led to some voxels showing different preferences in the main session compared to the localizer session *Peelen and Downing, 2005*).

After determining voxel preferences, we rearranged the seven task conditions according to each voxel's preference. The conditions are hereafter referred to as: P$^{at}$, P$^{at}$N, PN$^{at}$, N$^{at}$, P, PN, N, with *P* and *N* denoting the presence of the preferred and null stimuli, respectively, and the superscript *at* denoting the attended category. Mean voxel responses in the five ROIs for all task conditions are illustrated by navy lines in *Figure 3a–e*. Note that although the seven conditions constitute a discrete and not a continuous variable, we have connected the responses in attended conditions (in which body or house stimuli were attended) and unattended conditions (in which body and house were ignored and the fixation point color was attended) separately. This was done for visual purposes and ease of understanding.

We observed that the mean voxel response was generally higher when each stimulus was attended compared to the condition in which it was ignored. For instance, the response in the P$^{at}$ condition (in which the isolated preferred stimulus was attended) was higher than in the P condition (where the isolated preferred stimulus was ignored) in LO, pFs, and PPA ($ts > 3.6$, $ps < 0.01$, *corrected*), marginally higher in EBA ($t(18) = 2.69$, $p = 0.07$, *corrected*), and not significantly higher in V1 ($t(18) = 2.52$, $p = 0.1$, *corrected*). Similarly, comparing the N and N$^{at}$ conditions in each ROI, we observed an increase in response caused by attention in all ROIs ($ts > 4$, $ps < 0.01$, *corrected*) except for V1 ($t(18) = 2.4$, $p = 0.13$, *corrected*). A similar trend of response enhancement due to attention could also be observed in the paired conditions: attending to either stimulus increased the response in all ROIs ($ts > 4.4$, $ps < 0.01$, *corrected*) except for V1 ($ts < 2.59$, $ps > 0.08$, *corrected*). In all cases, the effect of attention was absent or only marginally significant in V1, which is not surprising since attentional effects are much weaker (*McAdams and Maunsell, 1999*) or even absent (*Luck et al., 1997*) in V1 compared to the higher-level regions of the occipito-temporal cortex. Next, we asked whether we could predict these response variations

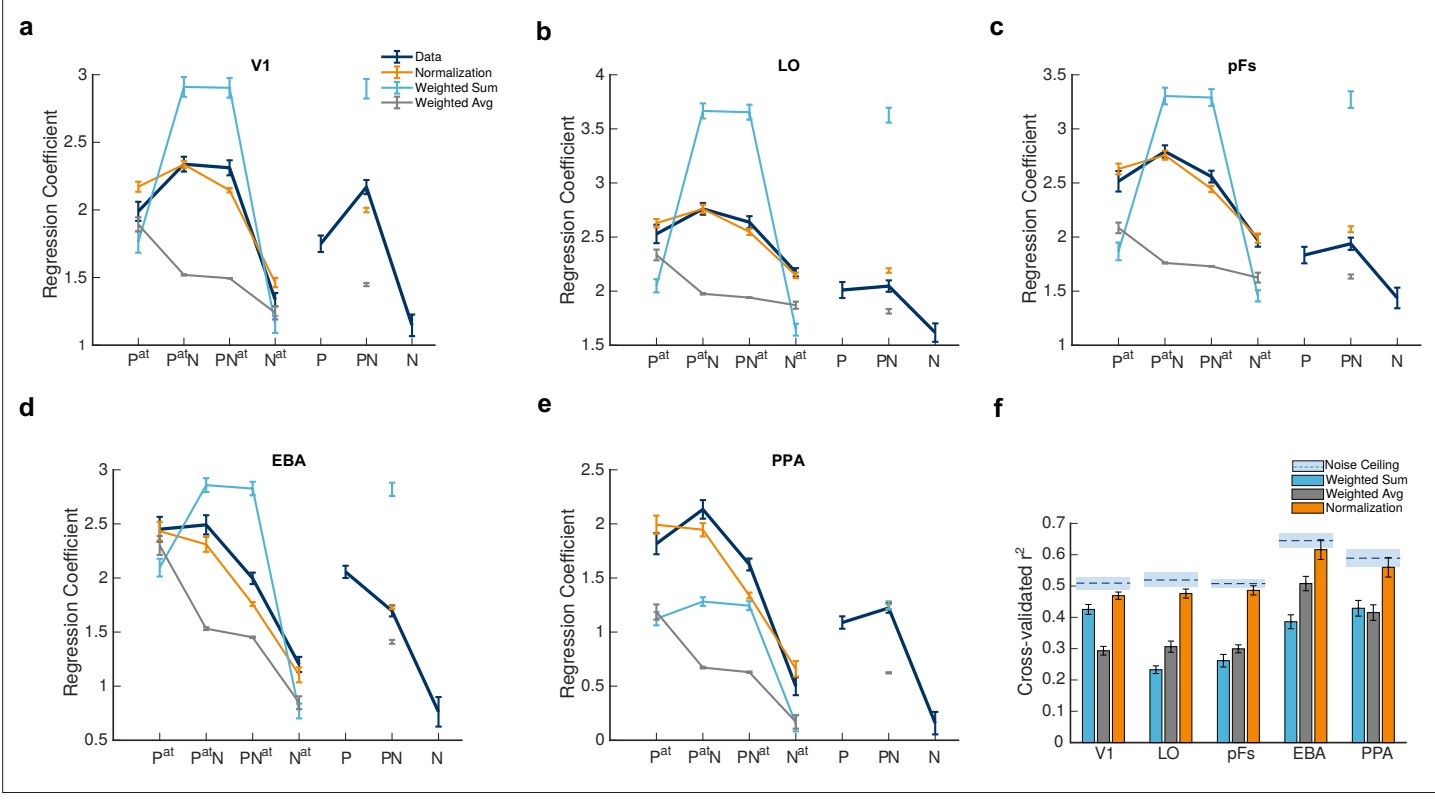

**Figure 3.** Divisive normalization explains voxel responses in different stimulus conditions. (**a–e**) Average fMRI responses and model predictions in the five regions of interest. Navy lines represent average responses. Light blue, gray, and orange lines show the predictions of the weighted sum, the weighted average, and the normalization models, respectively. The x-axis labels represent the 7 task conditions, $P^{at}$, $P^{at}N$, $PN^{at}$, $N^{at}$, P, PN, N, with P and N denoting the presence of the preferred and null stimuli and the superscript *at* denoting the attended category. For instance, P refers to the condition in which the unattended preferred stimulus was presented in isolation, and $P^{at}N$ refers to the paired condition with the attended preferred and unattended null stimuli. Error bars represent standard errors of the mean for each condition, calculated across participants after removing the overall between-subject variance. N = 19 human participants. (**f**) Mean explained variance, averaged over voxels in each region of interest for the 5 conditions predicted by the three models. Light blue, gray, and orange bars show the average variance explained by the weighted sum, the weighted average, and normalization models, respectively. Error bars represent the standard errors of the mean. N = 19 human participants. Dashed lines above each set of bars indicate the noise ceiling in each ROI, with the light blue shaded area representing the standard errors of the mean calculated across participants (see *Figure 3—figure supplement 1* for an example illustration of how the goodness of fit was calculated for each voxel). As observed in the figure, the normalization model was a better fit for the data compared to the weighted sum (ps < 0.02) and the weighted average (ps < 0.0001) models. Simulation results demonstrate that this superiority is not related to the higher number of parameters or the nonlinearity of the normalization model (see *Figure 3—figure supplement 2*).

The online version of this article includes the following figure supplement(s) for figure 3:

**Figure supplement 1.** Calculation of model goodness of fit for an example voxel.

**Figure supplement 2.** Goodness of fit of the three models for the simulated neural populations.

and attentional modulations caused by the change in the presented stimuli and the target of attention using three different models.

## Divisive normalization explains voxel responses in different stimulus conditions

We used the three models of weighted sum, weighted average, and normalization to predict voxel responses in different task conditions. Based on the weighted sum model, the response to multiple stimuli is determined by the sum of the responses to each individual stimulus presented in isolation, and attention to each stimulus increases the part of the response associated with the attended stimulus. For instance, in the presence of a null and a preferred stimulus with attention to the preferred stimulus, the response can be determined by $R_{P^{at},N} = \beta R_P + R_N$, with $R_{P^{at},N}$, $R_P$, and $R_N$, denoting the response elicited by both stimuli with attention directed to the preferred stimulus, the response to

the isolated preferred stimulus, and the response to the isolated null stimulus, respectively. $\beta$ is the attention-related parameter.

According to the weighted average model, the response to multiple stimuli is determined by the average of isolated-stimulus responses, and weighted by the parameter related to attention. Therefore, with an attended preferred and an ignored null stimulus, the response can be written as: $R_{P^{at},N} = \frac{\beta R_P + R_N}{2}$.

Finally, based on the normalization model, the response to a stimulus is determined based on the excitation due to that stimulus and the suppression due to the neighboring neuronal pool. Therefore, the response to an attended preferred and an ignored null stimulus is determined by: $R_{P^{at},N} = \frac{\beta c_P L_P + c_N L_N}{\beta c_P + c_N + \sigma}$, where $L_P$ and $L_N$ respectively denote the excitation caused by the preferred and the null stimulus, and $\sigma$ represents the semi-saturation constant. $c_P$ and $c_N$ are the respective contrasts of the preferred and null stimuli. Zero contrast for a stimulus denotes that the stimulus is not present in the visual field. In our experiment, we set contrast values to one when a stimulus was presented, and to zero when the stimulus was not presented (see Methods for detailed descriptions of models).

Although many studies have demonstrated that responses to multiple stimuli are added sublinearly in the visual cortex (*Heeger, 1992*; *Bloem and Ling, 2019*; *Reddy et al., 2009*; *Aqil et al., 2021*), it has been suggested that for weak stimuli, response summation can approach a linear or even a supralinear regime (*Rubin et al., 2015*; *Heuer and Britten, 2002*). Since the stimuli we used in this experiment were presented in a semi-transparent form and were therefore not in full contrast, we found it probable that the response might be closer to a linear summation regime in some cases. We thus used the weighted sum model to examine whether the response approaches linear summation in any region.

To compare the three models in their ability to predict the data, we split the fMRI data into two halves (odd and even runs) and estimated the model parameters separately for each voxel of each participant twice: once using the first half of the data, and a second time using the second half of the data. All comparisons of data with model predictions were made using the left-out half of the data in each case. All model results illustrate the average of these two cross-validated predictions. Note that this independent prediction is critical since the numbers of parameters in the three models are different. Possible over-fitting in the normalization model with more parameters will not affect the independent predictions (*Kay et al., 2013b*). The predictions of the three models for the five modeled task conditions are illustrated in *Figure 3a–e* (the two isolated ignored conditions P and N were excluded as they were used by the weighted sum and the weighted average models to predict responses in the remaining five conditions, see Methods).

As evident in the figure, the predictions of the normalization model (in orange) are generally better than the predictions of the weighted sum and the weighted average models (light blue and gray, respectively) in all regions. To quantify this observation, we calculated the goodness of fit for each voxel by taking the square of the correlation coefficient between the predicted model response and the respective fMRI responses across the five modeled conditions (*Figure 3—figure supplement 1*). We also calculated the noise ceiling in each region separately as the r-squared of the correlation between the odd and even halves of the data. Given that the correlation between the model and the data cannot exceed the reliability of the data (as calculated by the correlation between the data from odd and even runs), the r-squared can also not exceed the squared split-half reliability. The noise ceiling (squared split-half reliability), therefore, determines the highest possible goodness of fit a model can reach. The results are illustrated in *Figure 3f*.

We first compared the goodness of fit of the three models across the five ROIs using a $3 \times 5$ repeated measures ANOVA. The results showed a significant main effect of model ($F(2, 36) = 72.9$, $p < 0.0001$) and ROI ($F(4, 72) = 26.66$, $p < 0.0001$), and a significant model by ROI interaction ($F(8, 144) = 24.96$, $p < 0.0001$). On closer inspection, the normalization model was a better fit to the data than both the weighted sum ($ps < 0.02$, *corrected*) and the weighted average ($ps < 0.0001$, *corrected*) models in all ROIs. Since the normalization model had more parameters, we also used the AIC measure to correct for the difference in the number of parameters. The normalization model was a better fit according to the AIC measure as well (see *Supplementary file 2*). It is noteworthy that while the weighted average model performed better than the weighted sum model in LO and EBA ($ps < 0.002$, *corrected*), it was not significantly better in pFs and PPA ($ps > 0.37$, *corrected*), and worse than the weighted sum model in V1 ($p < 0.0001$, *corrected*).

We then calculated the normalization model's r-squared difference from the noise ceiling (NRD) for each ROI (*Equation 7*). NRD is a measure of the ability of the model in accounting for the explainable variation in the data; the lower the difference between the noise ceiling and a model's goodness of fit, the more successful that model is in predicting the data. We ran a one-way ANOVA to test for the effect of ROI on NRD, and observed that this measure was not significantly different across ROIs ($F(4,72) = 0.58$, $p = 0.61$), demonstrating that the normalization model was equally successful across ROIs in predicting the explainable variation in the data.

Interestingly, just focusing on the paired condition in which none of the stimuli were attended (the PN condition), the results of the weighted average model were closer to normalization (the gray and orange isolated data points on the subplots a-e of *Figure 3* are similarly close to the navy point of data in some regions). For this condition, the predictions of the normalization model were significantly closer to the data compared to the predictions of the weighted average model in V1, pFs, and PPA ($ps < 0.03$, *corrected*) but not significantly closer to the data in LO and EBA ($ps > 0.09$, *corrected*). These results are in agreement with previous studies suggesting that the weighted average model provides good predictions of neural and voxel responses in the absence of attention (*Zoccolan et al., 2005*; *Macevoy and Epstein, 2009*; *Kliger and Yovel, 2020*). However, when considering all the attended and unattended conditions, our results show that the normalization model is a generally better fit across all ROIs.

To ensure that the superiority of the normalization model over the weighted sum and weighted average models were not caused by the normalization model's nonlinearity or its higher number of parameters, we ran simulations of three neural populations. Neurons in each population calculated responses to multiple stimuli and attended stimuli by a summing, an averaging, and a normalizing rule (see Methods). We then used the three models to predict the population responses. Our simulation results demonstrate that despite the higher number of parameters, the normalization model is only a better fit for the population of normalizing neurons and not for summing or averaging neurons, as illustrated in *Figure 3—figure supplement 2*. These results confirm that the better fits of the normalization model cannot be related to the model's nonlinearity or its higher number of parameters.

## Normalization accounts for the change in response with the shift of attention

Next, comparing the responses in different conditions, we observed two features in the data. First, for the paired conditions, shifting attention from the preferred to the null stimulus caused a reduction in voxel responses. We calculated this reduction in response for each voxel by ($P^{at}N - PN^{at}$) (*Figure 4a*, top panel). This response change was significantly greater than zero in all ROIs ($ts > 6.2$, $ps < 0.0001$, *corrected*) except V1 ($t(18) = 0.66$, $p = 0.97$, *corrected*). Because the same stimuli were presented in both conditions but the attentional target changed from one category to the other, this change in response could only be related to the shift in attention and the stimulus preference of the voxels.

We then calculated the response change predicted by the three models to investigate model results in more detail. As illustrated in the bottom panel of *Figure 4a*, the orange bars depicting the predictions of the normalization model were very close to the navy bars depicting the observations in all ROIs, while the predictions of the weighted sum and the weighted average models (light blue and gray bars, respectively) were significantly different from the data in most regions.

To quantify this observation and to compare how closely the predictions of the three models followed the response change in the data, we calculated the difference between the response change observed in the data and the response change predicted by each model. Then, we ran a $3 \times 5$ repeated measures ANOVA with within-subject factors of the model and ROI on the obtained difference values. The results demonstrated a significant effect of model ($F(2,36) = 105.59$, $p < 0.0001$), a significant effect of ROI ($F(4,72) = 13.88$, $p < 0.0001$), and a significant model by ROI interaction ($F(8,144) = 28.63$, $p < 0.0001$). Post-hoc t-tests showed that the predictions of the normalization model were closer to the response change observed in the data in all ROIs ($ps < 0.0001$, *corrected*) except in V1, where the predictions of the weighted sum and the weighted average models were closer to the data ($ps < 0.0001$, *corrected*).

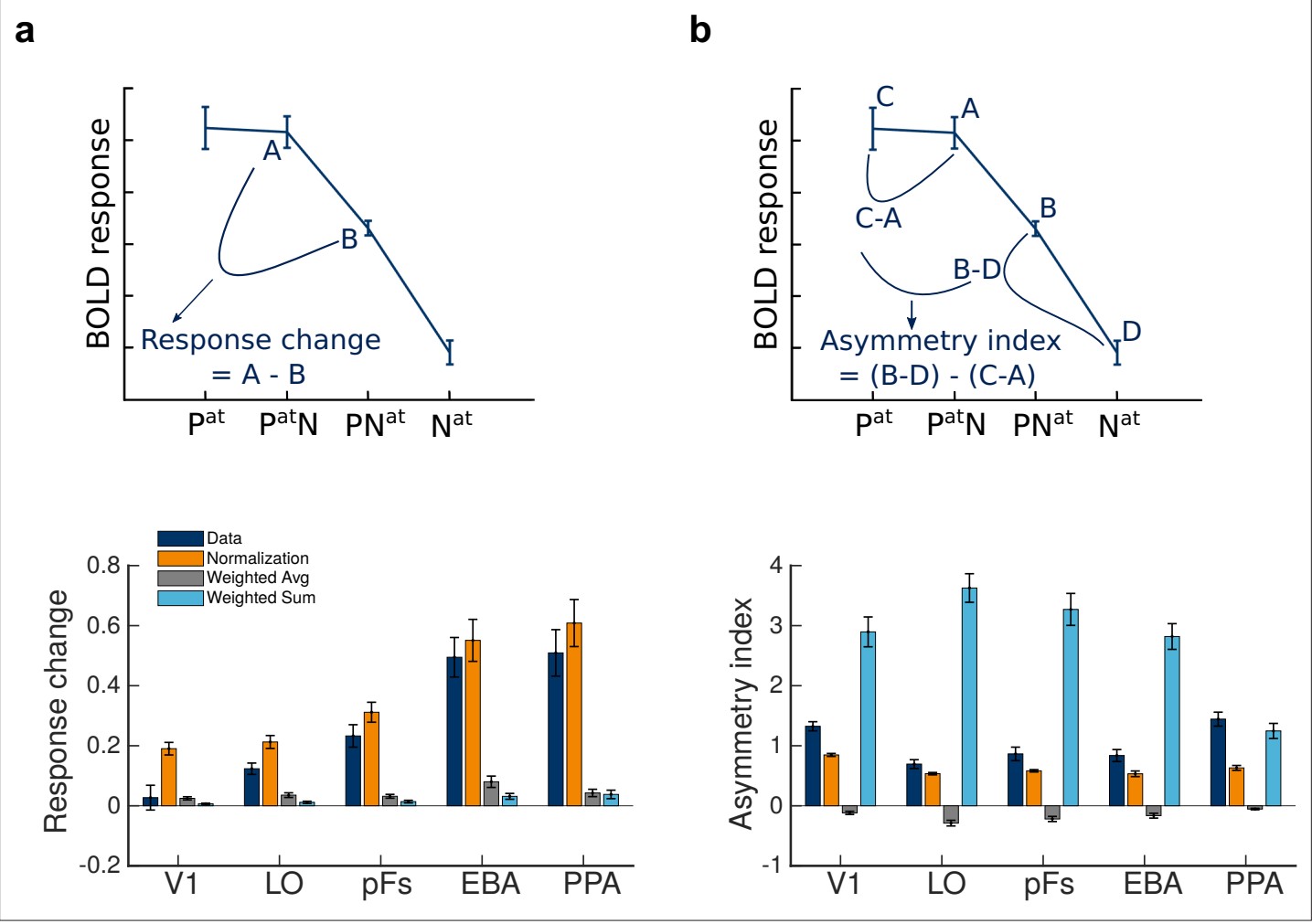

**Figure 4.** Normalization accounts for the observed effects of attention. (**a**) Top: Change in BOLD response when attention shifts from the preferred to the null stimulus in the presence of two stimuli, illustrated here for extrastriate body area (EBA). Bottom: The observed response change and the corresponding amount predicted by different models in different regions, calculated as illustrated in plot A. Error bars represent the standard errors of the mean. N = 19 human participants. (**b**) Top: The observed asymmetry in attentional modulation for attending to the preferred versus the null stimulus, depicted for EBA. Bottom: The observed and predicted asymmetries in attentional modulation in different regions, calculated as illustrated in plot B. Error bars represent the standard errors of the mean. N = 19 human participants.

### Asymmetry in attentional modulation is explained by the normalization model

The second feature we observed was that the effect of the unattended stimulus on the response depended on voxel selectivity for that stimulus, with the unattended preferred stimulus having larger effects on the response than the unattended null stimulus. Attending to the preferred stimulus in the presence of the null stimulus caused the response to approach the response elicited when attending to the isolated preferred stimulus. Therefore, attention effectively removed the effect of the null stimulus. However, attending to the null stimulus in the presence of the preferred stimulus did not eliminate the effect of the preferred stimulus and yielded a response well above the response elicited by attending to the isolated null stimulus. While this is the first time such asymmetry has been reported in human fMRI studies, these results are in agreement with previous monkey electrophysiology studies, showing the existence of an asymmetry in attentional modulation for attention to the preferred versus the null stimulus (*Lee and Maunsell, 2010*; *Ni et al., 2012*).

To quantify the observed asymmetry, we calculated an asymmetry index for each voxel by $(PN^{at} - N^{at}) - (P^{at} - P^{at}N)$, which is illustrated in the top panel of *Figure 4b*. This index was significantly greater than zero in all regions ($ts > 7.6$, $ps < 0.0001$, *corrected*).

Here, too, the normalization model was better at predicting the observed asymmetry in the data. The bottom panel of *Figure 4b* illustrates the asymmetry indices for the data and the three models in all regions. We calculated the difference between the asymmetry index observed in the data and the predicted index by each model and performed a $3 \times 5$ repeated measures ANOVA to compare the three models in how closely they predicted the asymmetry effect across ROIs using these difference values. We observed a significant effect of model ($F(2, 36) = 185.3$, $p < 0.0001$), a significant effect of ROI ($F(4, 72) = 64.97$, $p < 0.0001$), and a significant model by ROI interaction ($F(8, 144) = 45.60$, $p < 0.0001$). The prediction of the normalization model was closer to the data in all regions ($ps < 0.0001$, *corrected*) except for PPA, where the prediction of the weighted sum model was closer to the asymmetry observed in the data than the prediction of the normalization model ($p < 0.0001$, *corrected*).

## Other variants of the weighted average model

The weighted average model we used in previous sections had equal weights for the preferred and null stimuli, with attention biasing the attended preferred or null stimulus with the same amount (the weighted average EW model). However, different stimuli might have different weights in the paired response depending on the neurons' preference toward the stimuli. Besides, attention may bias preferred and null stimuli differently. Therefore, to examine the effect of unequal weights and attention parameters on the fits of the weighted average model, we tested two additional variants of this model.

To examine how unequal weights affect the fit of the weighted average model, we tested the weighted average UW model. Comparison of the fit of this model with the weighted average EW model showed that the UW variant was a significantly better fit than the EW model in all regions ($ts > 3.9$, $ps < 0.01$, *corrected*) except in LO, where it was a marginally better fit ($t(18) = 2.83$, $p = 0.054$, *corrected*). In the next step, to examine the effect of unequal weights and attention parameters on the fit of the weighted sum and the weighted average models, we tested the weighted average UWUB variant. This model had unequal weights and unequal attention parameters for the P and N stimuli. In this variant, no constraint was put on the sum of the weights. Thus, this model was effectively a generalization of the weighted sum and the weighted average models with four parameters. This model was a better fit than the weighted average EW model in all regions ($ts > 3.78$, $ps < 0.01$, *corrected*) except in EBA ($t(18) = 2.65$, $p = 0.08$, *corrected*).

We next compared the goodness of fit of all weighted average variants with the normalization model using a $4 \times 5$ ANOVA, as illustrated in *Figure 5a*. There was a significant effect of model ($F(3, 54) = 89.75$, $p < 0.0001$), a significant effect of ROI ($F(4, 72) = 34.97$, $p < 0.0001$), and a significant model by ROI interaction ($F(12, 216) = 7.55$, $p < 0.0001$). Post-hoc t-tests showed that these weighted average variants were still significantly worse fits to the data than the normalization model in all regions ($ps < 0.001$, *corrected*) except for EBA, where the normalization model was marginally better than the weighted average UWUB ($p = 0.065$, *corrected*).

Next, to examine whether the observed asymmetry was caused by response saturation, we tested a nonlinear variant of the weighted average model with saturation (the weighted average UWUB saturation model). This model's goodness of fit, as well as its predictions of asymmetry, are illustrated in *Figure 5b and c*. As illustrated in the figure, the saturation model's predicted asymmetry was closer to the data than normalization's prediction only in EBA ($p = 0.043$, *corrected*). In other regions, the normalization model's prediction of asymmetry was either significantly closer to the data (in V1, LO, and pFs, $ps < 0.0001$, *corrected*) or not significantly different from the saturation model (in PPA, $p = 0.63$, *corrected*). After running a $2 \times 5$ ANOVA to compare the fits of the normalization model and the weighted average UWUB saturation model across ROIs, we observed a significant effect of model ($F(1, 18) = 91.16$, $p < 0.0001$), a significant effect of ROI ($F(4, 72) = 19.46$, $p < 0.0001$), and a significant model by ROI interaction ($F(4, 72) = 4.82$, $p < 0.01$). Post-hoc t-tests showed that the normalization model was a significantly better fit than the saturation model in all ROIs ($ps < 0.01$, *corrected*).

## Discussion

Here, using single-voxel modeling, we examined the validity of the normalization model and demonstrated its superiority to the weighted sum and the weighted average models in predicting cortical

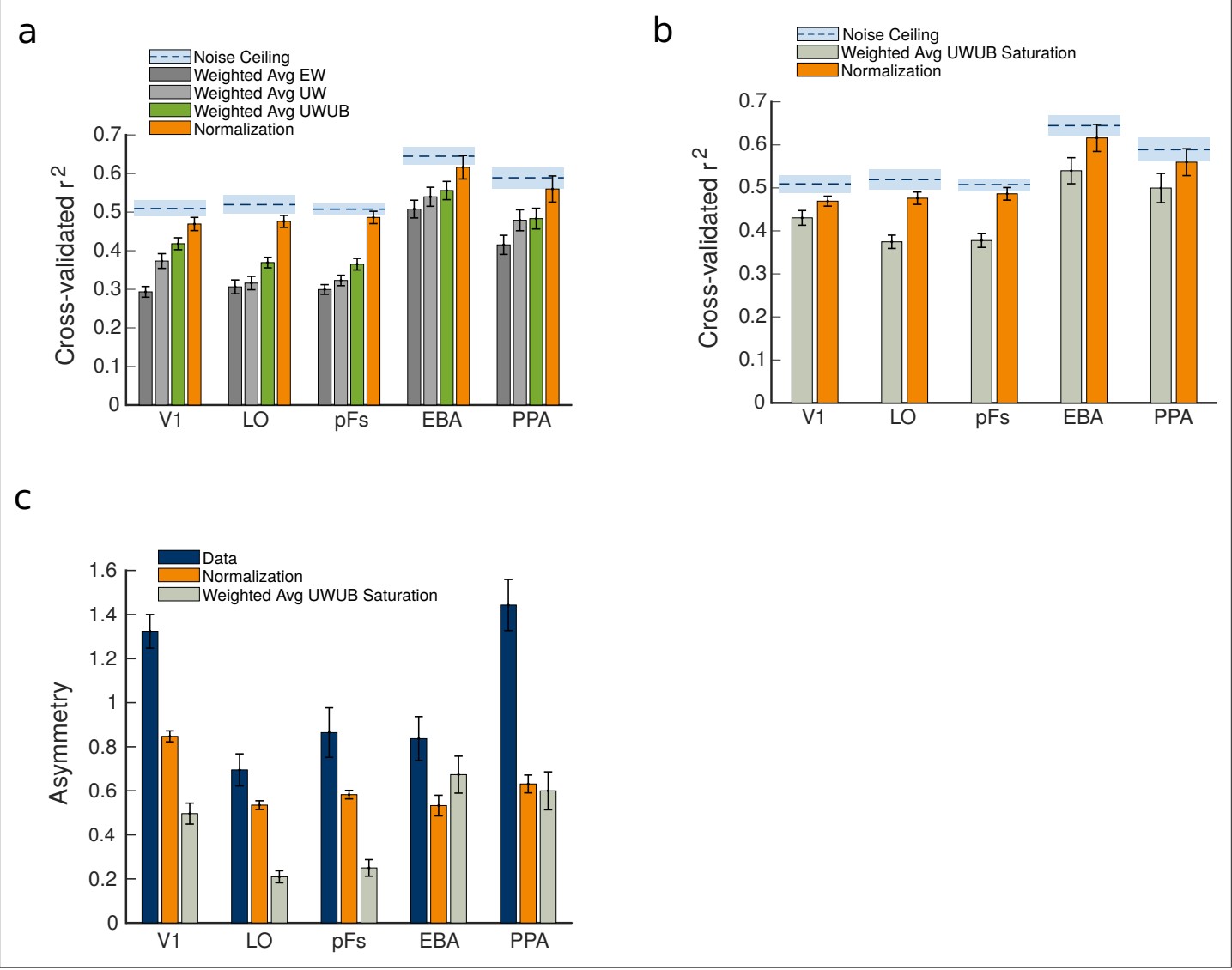

**Figure 5.** Comparison between the weighted average model variants and the normalization model predictions. (**a**) Comparison of the goodness of fit for weighted average variants and the normalization model. (**b**) The goodness of fit of the normalization model compared to the weighted average unequal weights and unequal betas (UWUB) saturation variant. (**c**) The asymmetry index was calculated for the data, compared to the predictions of the normalization model and the weighted average UWUB saturation model. Error bars represent the standard errors of the mean, calculated across N = 19 human participants.

responses to isolated and cluttered object stimuli. We also showed the success of the normalization model in predicting the observed effects of object-based attention, further suggesting it as a fundamental operation in the human brain.

While several electrophysiology studies have examined normalization in the monkey brain (*Bao and Tsao, 2018*; *Ni et al., 2012*; *Ni and Maunsell, 2017*; *Ni and Maunsell, 2019*), and although normalization has also been proposed to operate in the human brain (*Bloem and Ling, 2019*; *Kay et al., 2013a*), evidence for its validity in the human brain, particularly in the presence of attention, is still scarce. Expanding on the results of previous studies, showing the role of normalization in the human visual cortex for simple stimuli (*Kay et al., 2013a*; *Kay et al., 2013b*; *Aqil et al., 2021*), our work offers evidence for the role of normalization in multiple-object representation and object-based attention.

After comparing model predictions with the data, we investigated the effect of attention on multiple-object representation. Defining preferred and null stimuli for each voxel, we showed that

when attention shifted from the preferred to the null stimulus, there was a significant reduction in response in multiple regions of interest in the occipito-temporal cortex but not in the primary visual area. Furthermore, this response reduction increased as we moved to higher regions across the hierarchy, consistent with speculations of greater effects of top-down attention on higher regions of the visual cortex (*Cook and Maunsell, 2002*). Although this response reduction has also been predicted by the biased competition model (*Desimone and Duncan, 1995*), our results showed that the predictions of the weighted average model were significantly lower than the response reduction observed in the data in all ROIs except in V1. In contrast, the normalization model predicted the response reduction more accurately in all regions except in V1, where no significant response reduction was observed. Previous reports of the effects of attention in V1 have also been controversial, with some reporting attentional effects in this region (*Somers et al., 1999*; *Gandhi et al., 1999*) while others report little (*McAdams and Maunsell, 1999*) to no observed effects of attention (*Luck et al., 1997*). As suggested by *Heeger and Ress, 2002*, this might be due to the difference in task design and the experimental method used.

Moreover, our results indicated an asymmetry in attentional modulation when attending to the preferred versus the null stimulus. We demonstrated that while attention to a preferred stimulus almost eliminates the effect of the ignored null stimulus, attention to the null stimulus does not remove the effect of the preferred stimulus. Unlike response change with the shift of attention, which increases across the hierarchy, the asymmetry measured by our defined index remains approximately the same in all regions, indicating its dependence not on the top-down attentional signal but on the normalization computation performed in each region. This feature was also predicted by the normalization model but not by the weighted sum and the weighted average models.

Normalization has been reported to cause neural populations to operate in the averaging or winner-take-all regimes based on stimulus contrast (*Busse et al., 2009*). Here, we showed that in the presence of attention, responses can deviate from the averaging regime even without a change in contrast. We observed a winner-take-all behavior when the preferred stimulus was attended since its higher response along with its increase in gain due to attention, made it a much stronger input compared to the ignored null stimulus. On the other hand, when the null stimulus was attended, the response was closer to an average than a max-pooling response. This result explains why several previous studies in the object-selective regions indicated averaging as the rule for multiple-stimulus representation (*Zoccolan et al., 2005*; *Macevoy and Epstein, 2009*), while studies in regions with strong preferences for a particular category reported a winner-take-all mechanism (*Bao and Tsao, 2018*). We, therefore, extend previous reports of multiple-stimulus representation showing that the response is related to stimulus contrast (*Busse et al., 2009*) and neural selectivity (*Bao and Tsao, 2018*) by demonstrating that a combination of bottom-up and top-down signals act to yield a response that can be the average of the isolated responses, or a winner-take-all response, or somewhere between the two. We also demonstrate for the first time that the normalization model is superior to the weighted average model, which has often been used in lieu of the normalization model (*Zoccolan et al., 2005*; *Macevoy and Epstein, 2009*; *Kliger and Yovel, 2020*), in its ability to account for fMRI responses in the presence of attention. Testing other variants of the weighted average model with unequal weights and unequal attention parameters for the preferred and null stimuli, we demonstrate that the normalization model is a better fit compared to all these variants of the weighted average model.

Stimulus contrast has also been shown to have a crucial role in how single-stimulus responses are added to obtain the multiple-stimulus response. While responses to strong high-contrast stimuli are added sublinearly to yield the multiple-stimulus response, as predicted by the normalization model and the weighted average model, the sublinearity decreases for lower contrasts and even changes to linearity and supralinearity for weak stimuli (*Heuer and Britten, 2002*; *Rubin et al., 2015*). Here, since the stimuli we used were not in full contrast, we tested the weighted sum model as well to examine whether responses approach linearity in any region. Our results demonstrate that while the weighted average model generally performs better than the weighted sum model in the higher-level occipito-temporal cortex, the weighted sum model provides better predictions in V1. These results suggest stronger sublinearity in higher regions of the visual cortex compared to V1, which is in agreement with previous reports (*Kay et al., 2013b*). This observation might be related to the higher sensitivity of V1 neurons to contrast (*Goodyear and Menon, 1998*), causing a more significant decrease in V1

responses to low-contrast stimuli. This, in turn, might make the low-contrast stimulus weaker for V1 neurons, causing a move toward a lower level of sublinearity (*Sceniak et al., 1999*).

Attention to a stimulus has been suggested in the literature to be similar to an increase in the contrast of the attended stimulus (*Ni et al., 2012*), which is manifested in the similar effects of attention and contrast in the normalization equation. In this study, we presented the stimuli with a constant contrast but changed the number of stimuli and their attentional state to determine whether the normalization model could explain the effects of object-based attention in the human visual cortex, which has not been previously studied. We acknowledge that to fully ascertain the role of normalization in the human brain, we have to measure the contrast response function in each voxel to truly constrain the models and have conditions with varying levels of contrasts across attentional manipulations. Note that including variations of both attentional state and contrast is not trivial and is not possible with a single-session fMRI experiment. Our results remain suggestive of the role of normalization until these conditions are tested in future multi-session experiments.

Here, we compared the nonlinear normalization model with two linear models with fewer free parameters. To ensure that the difference in the number of free parameters did not affect the results, we used cross-validation and the AIC measure to compare model predictions with the data. If the success of the normalization model was due to its higher number of free parameters, it would affect its predictions for left-out data. We observed that the normalization model was also successful in predicting the left-out part of the data. In addition, we tested a nonlinear model variant with five free parameters. This model was still a worse fit than the normalization model. Finally, we used simulations of three different neural populations, with neurons in each population following either a summing, averaging, or normalization rule in their response to multiple stimuli and attended stimuli. Simulation results demonstrated that the normalization model was a better fit only for the normalizing population, confirming that the success of the normalization model is not due to its nonlinearity or the higher number of parameters but rather as a result of it being a closer estimation of the computation performed at the neural level for object representation.

It is noteworthy that here, we are looking at the BOLD responses. We are aware of the limitations of the fMRI technique as the BOLD response is an indirect measure of the activity of neural populations. While an increase in the BOLD signal could be related to an increase in the neuronal firing rates of the local population (*Logothetis et al., 2001*), it could also be related to subthreshold activity resulting from feedback from the downstream regions of the visual cortex (*Heeger and Ress, 2002*). The observed effects, therefore, may be related to local population responses or may be influenced by feedback from downstream regions. Also, since the measured BOLD signal is related to the average activity of the local population, and we do not have access to single-unit responses, some effects may change in the averaging process. Nevertheless, our simulation results show that the effects of the normalization computation are preserved even after averaging. We should keep in mind, though, that these are only simulations and are not based on data directly measured from neurons. Future experiments with intracranial recordings from neurons in the human visual cortex would be invaluable in validating our results.

Another limitation in interpreting the results is related to a possible stronger saturation of the BOLD response, which can potentially explain the observed asymmetry in attentional modulation. Since the asymmetry in attentional modulation has also been previously reported at the neural level (*Ni et al., 2012*), this effect is unlikely to be exclusively caused by the saturation in the BOLD signal. It is noteworthy, however, that saturation is a characteristic of cortical responses even at the neural level. Whether this effect at the neural level is caused by response saturation or as a result of the normalization computation cannot be distinguished from our current knowledge. Nevertheless, we tested a variant of the weighted average model with an extra saturation parameter. Although this model could partially predict the observed asymmetry, the predictions were worse than the normalization model's predictions. Also, this model was an overall worse fit to the data compared to the normalization model. The normalization model, therefore, provides a more parsimonious account of the data. Nevertheless, we have only tested a saturation model. The exact nonlinearities that affect the transformation of neural population responses to the BOLD response are not fully mapped out yet, especially in cases where multiple overlapping stimuli are presented in the visual scene. It is possible that the advantage of the normalization model could be at least partly related to these nonlinearities. Modeling such nonlinearities requires experiments that simultaneously record fMRI and neural data.

The validity of our conclusions about the superiority of the normalization model should be reevaluated after such data becomes available.

In sum, our results indicate that the normalization model predicts responses at the voxel level beyond the primary visual cortex and across the visual hierarchy, especially in higher-level regions of the human occipito-temporal cortex, with and without attention and in conditions with isolated or cluttered stimuli. We, therefore, provide evidence suggesting divisive normalization as a canonical computation operating in the human brain during object-based attention.

## Methods

### Participants

A total of 21 healthy right-handed participants (10 females, 20–40, all with normal or corrected-to-normal vision) participated in the experiment. All participants gave written consent prior to their participation and received payment for their participation in the experiment. Imaging was performed according to safety guidelines approved by the ethics committee of the Institute for Research in Fundamental Sciences (IPM). Data from two participants were removed from the analysis because of excessive head motion (more than 3 mm throughout the session). This exclusion criteria was established before running the experiment. Based on previous studies with sample sizes of 5–10 (*Bloem and Ling, 2019*; *Reddy et al., 2009*; *Kay et al., 2013a*; *Kay et al., 2013b*), we are confident that with this sample size, we have enough power for comparing different model predictions of fMRI responses.

### Stimuli and experimental procedure

Stimuli were from the two categories of human bodies and houses similar to the ones used in previous studies (*Vaziri-Pashkam and Xu, 2017*; *Vaziri-Pashkam and Xu, 2019*; *Xu and Vaziri-Pashkam, 2019*). Each category consisted of ten unique exemplars in gray-scale format (*Figure 1a*). These exemplars differed in identity, pose (for bodies), and viewing angle (for houses). Stimuli were fitted into a transparent square subtending 10.2° of visual angle and placed on a gray background. A central red fixation point subtending 0.45° of visual angle was present throughout the run. Stimuli from each category were presented in semi-transparent form, in isolation, or paired with stimuli from the other category.

In a blocked-design paradigm, participants were instructed by a word cue to attend to bodies, houses, or the color of the fixation point at the beginning of each block. Therefore, the stimuli from each category were either attended (when the category was cued), or ignored (when the fixation point color was cued), in isolation or cluttered by stimuli from the other category. The main experiment thus consisted of seven blocks with all possible combinations of stimulus categories and attention: Attend isolated bodies, Attend cluttered bodies, Attend isolated houses, Attend cluttered houses, Ignore isolated bodies, Ignore isolated houses, Ignore cluttered bodies and houses (*Figure 1c*). In blocks with attention to bodies or houses, participants performed a one-back repetition detection task on the attended stimuli by pressing a button when the exact same stimulus appeared in two consecutive trials. In blocks with attention directed toward the fixation point color, participants responded when the color of the fixation point changed from red to orange. These blocks served as conditions in which the visual stimuli (bodies and houses) were ignored. Target repetition and fixation point color change occurred 2–3 times at random in each block. There was also an additional fixation color block in each run, with a red fixation point presented in the middle of the gray screen and in the absence of stimuli from either category. The participants' task in this block was to detect a fixation point color change. The fixation point color changed to orange two or three times during the block. We used a contrast between the BH condition (with the same task but with the presence of both body and house stimuli) and this fixation condition to select the voxels in each ROI that were responsive to the presented stimuli. This was especially important for V1 voxels, since the stimuli presented in the early visual area localizer were larger than the stimuli presented in the main experiment.

Each run started with an 8 s fixation. Each block lasted for 10 s, starting with a 1 s cue and a 1 s fixation. 10 exemplars from one or both categories were presented during the block, each for 400 ms, followed by 400 ms of fixation. There was an 8 s fixation between blocks, and a final 8 s fixation at the end of the last block. The presentation order of the blocks was random, and counterbalanced

across the experimental runs. Each run lasted 2 min 32 s. For the main experiment, 14 participants completed 16 runs, two participants completed 14 runs, and three participants completed 12 runs.

## Localizer experiments

In this study we examined the primary visual cortex V1 along with the object-selective areas LO and pFs, the body-selective EBA, and the scene-selective PPA. All participants completed four localizer runs which were used to define the primary visual and the category-selective ROIs. We used previously established protocols for the localizer experiments, but the details are repeated here for clarification and convenience.

### Early visual area localizer

We used meridian mapping to localize the primary visual cortex V1. Participants viewed a black-and-white checkerboard pattern with a diameter of 27.1° of visual angle through a 60 degree polar angle wedge aperture. The wedge was presented either horizontally or vertically. Participants were asked to detect a luminance change in the wedge in a blocked-design paradigm. Each run consisted of four horizontal and four vertical blocks, each lasting 16 s, with 16 s of fixation in between. A final 16 s fixation followed the last block. Each run lasted 272 s. The order of the blocks was counterbalanced within each run. Each participant completed two runs of this localizer.

### Category localizer

A category localizer was used to localize the cortical regions selective to scenes, bodies, and objects. In a blocked-design paradigm, participants viewed stimuli from the five categories of faces, scenes, objects, bodies, and scrambled images, with each stimulus subtending 14.3° of visual angle. Each localizer run contained two 16 s blocks of each category, with the presentation order counterbalanced within each run. An 8 s fixation period was presented at the beginning, middle, and end of the run. In each block, 20 stimuli from the same category were presented. Each trial lasted 750 ms with 50 ms fixation in between. Participants were asked to maintain their fixation on a red circle at the center of the screen and press a key when they detected a slight jitter in the stimuli. Participants completed two runs of this localizer, each lasting 344 s. LO (*Malach et al., 1995*), pFs (*Grill-Spector et al., 1998*), EBA (*Downing et al., 2001*), and PPA (*Epstein et al., 1999*) were then defined using this category localizer.

## Data acquisition

Data were collected on a Siemens Prisma MRI system using a 64-channel head coil at the National Brain-mapping Laboratory (NBML). For each participant, we performed a whole-brain anatomical scan using a T1-weighted MP-RAGE sequence at the beginning of data acquisition. For the functional scans, including the main experiment and the localizer experiments, 33 slices parallel to the AC-PC line were acquired using T2*-weighted gradient-echo echo-planar sequences covering the whole brain (TR = 2 s, TE = 30 ms, flip angle = 90°, voxel size = 3 × 3 × 3 mm³, matrix size = 64 × 64). The stimuli were back-projected onto a screen using an LCD projector with the refresh rate of 60 Hz and the spatial resolution of 768 × 1024 positioned at the rear of the magnet, and participants observed the screen through a mirror attached to the head coil. MATLAB and Psychtoolbox were used to create all stimuli.

## Preprocessing

fMRI data analysis was performed using FreeSurfer (https://surfer.nmr.mgh.harvard.edu) and in-house MATLAB codes. fMRI data preprocessing steps included 3D motion correction, slice timing correction, and linear and quadratic trend removal. The data in each run were motion-corrected per-run and aligned to the anatomical data using the middle time point of that run. The fMRI data from the localizer was smoothed using a 5 mm FWHM Gaussian kernel, but no spatial smoothing was performed on the data from the main experiment to optimize the voxel-wise analyses. A double gamma function was used to model the hemodynamic response function. We eliminated the first four volumes of each run to allow the signal to reach a steady state.

## ROI definition

Using freesurfer's tksurfer module, we determined the primary visual cortex V1 using a contrast of horizontal versus vertical polar angle wedges to reveal the topographic maps in the occipital cortex (*Sereno et al., 1995*; *Tootell et al., 1998*). To define the object-selective areas LO in the lateral occipital cortex and pFs in the posterior fusiform gyrus (*Malach et al., 1995*; *Grill-Spector et al., 1998*), we used a contrast of objects versus scrambled images. Active voxels in the lateral occipital and ventral occipitotemporal cortex were selected as LO and pFS, respectively, following the procedure described by *Kourtzi and Kanwisher, 2000*. We used a contrast of scenes versus objects for defining the scene-selective area PPA in the parahippocampal gyrus (*Epstein et al., 1999*), and a contrast of bodies versus objects for defining the body-selective area EBA in the lateral occipitotemporal cortex (*Downing et al., 2001*). The activation maps for both localizers were thresholded at $p < 0.001$, uncorrected.

## Data analysis

We performed a general linear model (GLM) analysis for each participant to estimate voxel-wise regression coefficients for each of the 8 task conditions. The onset and duration of each block were convolved with a hemodynamic response function and were entered to the GLM as regressors. Movement parameters and linear and quadratic nuisance regressors were also included in the GLM. We then used these obtained coefficients to compare the BOLD response in different conditions in each ROI. In order to compensate for the difference in size between the localizer stimuli and the stimuli presented in the main experiment, we selected active voxels in each ROI using a contrast between the BH condition (with ignored body and house stimuli) and the fixation block (with no stimuli presented). We selected the voxels that were significantly active during the BH condition compared to the fixation block (with $p < 0.01$) across all runs for any further analyses. Preferred and null categories for each voxel were determined using voxel responses in conditions with isolated stimuli with the participant performing the task on the fixation point (color blocks). We then determined the activity in seven conditions for each voxel: $P^{at}$, $P^{at}N$, $PN^{at}$, $N^{at}$, P, PN, N, with P and N denoting the presence of the preferred and null stimuli and the superscript *at* denoting the attended category. For instance, P refers to the condition in which the unattended preferred stimulus was presented in isolation, and $P^{at}N$ refers to the paired condition with the attended preferred and unattended null stimuli.

### Model details

We used three models to predict the results: the weighted sum, the weighted average, and the normalization models. The weighted sum model is a simple linear model suggesting that the response to multiple stimuli is the sum of responses to individual stimuli, and attention to a stimulus increases the response to that stimulus by the attention-related parameter, $\beta$:

$$R_{P,N} = R_P + R_N \tag{1a}$$

$$R_{P^{at},N} = \beta R_P + R_N \tag{1b}$$

$$R_{P,N^{at}} = R_P + \beta R_N \tag{1c}$$

, with $R_{P,N}$ denoting the response elicited with the preferred and null stimuli present in the receptive field, and $R_P$ and $R_N$ denoting the response to isolated preferred and null stimuli, respectively. The superscript *at* specifies the attended stimulus, and the stimulus is ignored otherwise.

The weighted average model (*Zoccolan et al., 2005*; *Macevoy and Epstein, 2009*; *Baeck et al., 2013*) is also a linear model that posits the response to multiple stimuli as the average of individual responses. Similar to the weighted sum model, the response of an attended stimulus is enhanced by the parameter related to attention, $\beta$:

$$R_{P,N} = \frac{R_P + R_N}{2} \tag{2a}$$

$$R_{P^{at},N} = \frac{\beta R_P + R_N}{2} \tag{2b}$$

$$R_{P,N^{at}} = \frac{R_P + \beta R_N}{2} \tag{2c}$$

The normalization model of attention (*Heeger, 1992*; *Carandini et al., 1997*; *Reynolds and Heeger, 2009*; *Ni et al., 2012*) can be described using divisive normalization with a saturation term in the denominator:

$$R_{P,N} = \frac{c_P L_P + c_N L_N}{c_P + c_N + \sigma} \tag{3}$$

Here, $L_P$ and $L_N$ denote the excitatory drive induced by the preferred or the null stimulus, respectively and $\sigma$ represents the semi-saturation constant. $c_P$ and $c_N$ are the respective contrasts of the stimuli. Zero contrast denotes that the respective stimulus is not present in the visual field. In our experiment, we set contrast values to one when a stimulus was presented and to zero when the stimulus was not presented. When attention is directed towards one of the stimuli, we can rewrite the *Equation 3* as:

$$R_{P,N} = \frac{c_P L_P + c_N L_N}{c_P + c_N + \sigma} \tag{3a}$$

$$R_{P^{at},N} = \frac{\beta c_P L_P + c_N L_N}{\beta c_P + c_N + \sigma} \tag{3b}$$

$$R_{P,N^{at}} = \frac{c_P L_P + \beta c_N L_N}{c_P + \beta c_N + \sigma} \tag{3c}$$

It is noteworthy that the normalization model is different in nature from the two linear models and takes into account the suppression caused by the neighboring pool even in the presence of a single stimulus. We cannot, therefore, use the measured response in isolated conditions as the excitatory drive in paired conditions. Rather, we need extra parameters to estimate the excitation caused by each stimulus. We then use this excitation to predict the response to attended and ignored stimuli in isolated and paired conditions (*Ni et al., 2012*; *Ni and Maunsell, 2019*). On the other hand, the weighted sum and the weighted average models are not concerned with the underlying excitation and suppression. The assumption of these models is based on the resulting response that we actually measure in the paired condition, considering it to be respectively the sum or the average of the measured response in the isolated conditions. In order to take the difference in the number of model parameters into account, we have used both cross-validated r-squared on independent data and AIC measures (see the section on model comparison).

We fit model parameters for the three models. $\beta$ was fit as a free parameter for all models. The normalization model had three additional free parameters, $L_P$, $L_N$, and σ. σ and $\beta$ were constrained to be greater than zero and one, respectively, and less than 10. $L_P$ and $L_N$ were constrained to have an absolute value of less than 10. We estimated model parameters using constrained nonlinear optimizing, which minimized the sum-of-square errors. Values of the estimated parameters of the weighted sum, weighted average, and normalization models are provided in *Supplementary files 3–5*, respectively, for the odd and even runs.

## Weighted average model variants

In addition to the three main models, we tested three variants of the weighted average model. The main weighted average model had equal weights for the two stimuli (weighted average EW). For the first variant, we examined a weighted average model with unequal weights for the two stimuli (weighted average UW). According to this model, the response to two simultaneously-presented stimuli was a weighted average of the responses to isolated stimuli, but in contrast to the main weighted average model we used, the weights were not equal in this average. Instead, each stimulus had a different weight in the average, but the sum of the weights was set to 1:

$$R_{P,N} = \alpha R_P + (1 - \alpha) R_N \tag{4a}$$

$$R_{P^{at},N} = \beta \alpha R_P + (1 - \alpha) R_N \tag{4b}$$

$$R_{P,N^{at}} = \alpha R_P + \beta (1 - \alpha) R_N \tag{4c}$$

Here, $R_{P,N}$ denotes the response elicited with the preferred and null stimuli present in the receptive field, and $R_P$ and $R_N$ denote the response to isolated preferred and null stimuli, respectively. The superscript *at* specifies the attended stimulus, and the stimulus is ignored otherwise. $\alpha$ denotes the weight of the preferred stimulus, and $\beta$ is the attention-related parameter.

The second variant we tested was a weighted average variant with unequal weights and unequal betas (weighted average UWUB). Based on this model, the preferred and null stimuli had different weights and different attention-related parameters:

$$R_{P,N} = \alpha_P R_P + \alpha_N R_N \tag{5a}$$

$$R_{P^{at},N} = \beta_P \alpha_P R_P + \alpha_N R_N \tag{5b}$$

$$R_{P,N^{at}} = \alpha_P R_P + \beta_N \alpha_N R_N \tag{5c}$$

, where $\alpha_P$ and $\alpha_N$ respectively denote the weight of the preferred and null stimuli, and $\beta_P$ and $\beta_N$ denote the attention-related parameter of the preferred and null stimuli, respectively. Here, there was no limitation on the sum of weights to equal 1, as was the case for the weighted average EW and weighted average UW models. Therefore, this model was a generalization of the weighted sum and weighted average models.

Lastly, we tested a nonlinear variant with unequal weights, unequal betas, and an extra saturation parameter (weighted average UWUB saturation). Being similar to the weighted average UWUB model in theory, it also estimated a saturation value, s, for each voxel. After parameter estimation, the minimum value of the calculated response and the estimated saturation parameter, $s$, was chosen as the response:

$$R_{P,N} = min((\alpha_P R_P + \alpha_N R_N), s) \tag{6a}$$

$$R_{P^{at},N} = min((\beta_P \alpha_P R_P + \alpha_N R_N), s) \tag{6b}$$

$$R_{P,N^{at}} = min((\alpha_P R_P + \beta_N \alpha_N R_N), s) \tag{6c}$$

Values of the estimated parameters of the weighted average UW, UWUB, and UWUB saturation model variants are provided in *Supplementary files 6–8*, respectively, for the odd and even runs.

## Model-data comparison

We split the fMRI data into two halves of odd and even runs and estimated model parameters for the first half as described. Then, using the estimated parameters for the first half, we calculated model predictions for each voxel in each condition and compared the predictions with the left-out half of the data. All comparisons of data with models, including the calculation of the goodness of fit, were done using the left-out data. We repeated this procedure twice: once using the odd half of the data for parameter estimation and the even half for comparing model predictions with the data, and a second time using the even half of the data for parameter estimation and the odd half for comparison with the model predictions. All figures, including model results, illustrate the average of the two repetitions. Since the weighted sum and the weighted average models used the response in the P and N conditions to predict responses in the remaining five conditions, we only used these five conditions and excluded the P and N conditions when calculating the goodness of fit for all models. The goodness of fit was calculated by taking the square of the correlation coefficient between the observed and predicted responses for each voxel across the five modeled conditions (*Figure 3—figure supplement 1*). We also calculated the correlation between voxel responses of the two halves of the data across the same five conditions and calculated the noise ceiling in each ROI as the squared coefficient of this correlation. We determined the r-squared difference from the noise ceiling (NRD) in each ROI by calculating the difference between the noise ceiling and the model's goodness of fit in that ROI:

$$NRD = Noise\ ceiling - Goodness\ of\ fit \tag{7}$$

We compared the goodness of fit of the three models across all ROIs using a repeated measures $3 \times 5$ ANOVA (see the statistics section). We also compared the NRD of the normalization model across all ROIs using a one-way ANOVA. In order to compensate for the difference in the number of parameters for different models, we used the Akaike Information Criterion (AIC) (*Burnham and Anderson, 2004*; *Denison et al., 2021*). Under the assumption of a normal distribution of error, AIC is calculated by:

$$AIC = n\ ln(\frac{RSS}{n}) + 2k + C \tag{8}$$

, where n denotes the number of observations, RSS denotes the residual sum of squares, k is the number of free parameters of the model, and C is a constant with the same amount for all models.

A smaller AIC value shows that the model fits the data better. We therefore calculated ΔAIC for all model pairs.

## Simulation

To further check whether the success of the normalization model was due to its higher number of parameters or as a result of it being a closer estimation of the performed neural computations, we used a simulation approach. We simulated neural responses for single and multiple stimuli in the absence and presence of attention. In a neural population composed of $10^4$ neurons, neurons were body- or house-selective. Each neuron also responded to the category other than its preferred category, but to a lesser degree and with variation. We had three kinds of neurons: (i) summing neurons, for which the response to multiple stimuli and attended stimuli was calculated based on the weighted sum model, (ii) averaging neurons, which behaved based on the weighted average model, and (iii) normalizing neurons, which behaved based on divisive normalization. We chose neural responses and the attention factor randomly from a range comparable with neural studies of attention and object recognition in the ventral visual cortex (*Ni et al., 2012*; *Bao and Tsao, 2018*). Using equations discussed for each of the models, we calculated the response of each neuron to the seven conditions of our main fMRI experiment. Then, we randomly chose 200 neurons from the population, with the ratio of body/house preference similar to each of the ROIs in the main experiment. We then averaged the selected neurons' responses to make up a voxel and added Gaussian noise to the voxel, 16 times for each voxel with a different Gaussian noise every time to make up 16 measurements (16 runs, as in the fMRI experiment) for each condition. We had 30 voxels for each ROI. Then, dividing the runs into two halves, we performed the same modeling process as in the fMRI experiment. For the three models of weighted sum, weighted average, and normalization, we estimated model parameters for one-half of the data and predicted voxel responses for the second half of the data.

## Quantifying attentional effects

We defined two indices to quantify the observed effects of attention. The first index was used to compare voxel activities in paired conditions in which attention was directed toward the objects. We defined the response change index as the difference in average voxel activity when attention shifted from the preferred to the null stimulus, $(P^{at}N - PN^{at})$. The second index was used to quantify the asymmetry in attentional modulation. The asymmetry index, $[(PN^{at} - N^{at}) - (P^{at} - P^{at}N)]$, compared the effect of the unattended stimulus on the response in conditions with unattended preferred or null stimuli. The comparison of observed indices with indices calculated from model predictions was done using the left-out part of the data.

## Statistics

We performed sets of repeated measures ANOVAs to test for the main effects of the model, ROI, and their interaction for model goodness of fit and the reported effects of attention. For all performed ANOVAs, we used the Mauchly's test to check whether the assumption of sphericity had been met. For cases where the assumption of sphericity had been violated, we used the Greenhouse-Geisser estimate to correct the degrees of freedom. Where applicable, we corrected for multiple comparisons using the Dunn-Sidak procedure.

To compare the goodness of fit of the three models across ROIs, we performed a $3 \times 5$ repeated measures ANOVA with within-subject factors of the model (weighted sum, weighted average, and normalization), and ROI (V1, LO, pFs, EBA, and PPA) to test for the main effects of model and ROI and their interaction. Mauchly's test indicated that the assumption of sphericity had been violated ($p < 0.001$). We thus corrected the degrees of freedom using the Greenhouse-Geisser estimate ($\epsilon = 0.22$). We also ran a one-way ANOVA to test for the effect of ROI on the difference between the noise ceiling and the normalization model's r-squared (*Equation 7*). Mauchly's test indicated that the assumption of sphericity had been violated ($p = 0.014$). The degrees of freedom were corrected using the Greenhouse-Geisser estimate ($\epsilon = 0.69$).

We then calculated the difference between the observed effects of attention in the data and the predictions of each model. To compare model predictions of the two attentional effects across ROIs, we ran two sets of $3 \times 5$ repeated measures ANOVAs with within-subject factors of model and ROI. Mauchly's test indicated that the assumption of sphericity was met for both tests.

To compare the weighted average UW and UWUB model variants with the weighted average EW model and the normalization model, we ran a 4 × 5 repeated measures ANOVA with within-subject factors of model and ROI. Mauchly's test indicated that the assumption of sphericity had been violated ($p < 0.001$), so we used the Greenhouse-Geisser estimate to correct the degrees of freedom ($\epsilon = 0.19$). Finally, to compare the fits of the normalization model and the weighted average UWUB saturation model across ROIs, we ran a 2 × 5 repeated measures ANOVA. Mauchly's test showed a violation of the assumption of sphericity ($p < 0.001$), so the Greenhouse-Geisser estimate was used to correct the degrees of freedom ($\epsilon = 0.37$).

## Acknowledgements

Maryam Vaziri-Pashkam was supported by NIH Intramural Research Program ZIA-MH002035.

## Additional information

### Funding

| Funder | Grant reference number | Author |
| --- | --- | --- |
| National Institutes of Health | ZIA-MH002035 | Maryam Vaziri-Pashkam |

The funders had no role in study design, data collection and interpretation, or the decision to submit the work for publication.

### Author contributions

Narges Doostani, Conceptualization, Data curation, Formal analysis, Investigation, Visualization, Writing - original draft, Writing – review and editing; Gholam-Ali Hossein-Zadeh, Resources, Supervision, Writing – review and editing; Maryam Vaziri-Pashkam, Conceptualization, Supervision, Writing – review and editing

### Author ORCIDs

Narges Doostani  https://orcid.org/0000-0001-5775-6595
Maryam Vaziri-Pashkam  https://orcid.org/0000-0003-1830-2501

### Ethics

All participants gave written consent prior to their participation in the experiment. Imaging was performed according to safety guidelines approved by the ethics committee of the Institute for Research in Fundamental Sciences with the reference number 98/60.1/2184.

### Decision letter and Author response

Decision letter https://doi.org/10.7554/eLife.75726.sa1
Author response https://doi.org/10.7554/eLife.75726.sa2

## Additional files

### Supplementary files

- Supplementary file 1. Voxel preference consistency percentage across odd and even runs.
- Supplementary file 2. Difference between the AIC values of the fits of the three models.
- Supplementary file 3. Estimated parameters of the weighted sum model.
- Supplementary file 4. Estimated parameters of the weighted average model.
- Supplementary file 5. Estimated parameters of the normalization model.
- Supplementary file 6. Estimated parameters of the weighted average UW model.
- Supplementary file 7. Estimated parameters of the weighted average UWUB model.
- Supplementary file 8. Estimated parameters of the weighted average UWUB saturation model.
- MDAR checklist

## Data availability

fMRI data have been deposited in OSF under DOI https://doi.org/10.17605/OSF.IO/8CH9Q.

The following previously published dataset was used:

| Author(s) | Year | Dataset title | Dataset URL | Database and Identifier |
|---|---|---|---|---|
| Doostani N, Vaziri-Pashkam M, Hossein-Zadeh GH | 2021 | Normalization_Task | https://doi.org/10.17605/OSF.IO/8CH9Q | Open Science Framework, 10.17605/OSF.IO/8CH9Q |

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
