## [Editor Report]

This study on object-based attention furthers of understanding of the role of normalization across the visual hierarchy in the human visual cortex. The authors provide solid functional MRI evidence that supports their claims, demonstrating that the normalization model predicts the observed effect when participants selectively attend to one of two stimulus categories. The paper is an important contribution to the fields of perceptual and cognitive neuroscience.

---

## [Decision Letter]

**Decision letter after peer review:**

Thank you for submitting your article "Evidence for Normalization as a Fundamental Operation Across the Human Visual Cortex" for consideration by *eLife*. Your article has been reviewed by 3 peer reviewers, one of whom is a member of our Board of Reviewing Editors, and the evaluation has been overseen by Tirin Moore as the Senior Editor. The reviewers have opted to remain anonymous.

The reviewers have discussed their reviews with one another, and the Reviewing Editor has drafted this to help you prepare a revised submission. Please note that the reviewers think that the paper has potential but extensive revisions are needed.

Essential Revisions:

Theoretical framework and interpretation

(1) The title is misleading and a bit grandiose – "In" instead of "across". Across would imply evidence for normalization across all ~30 areas of visual brain, whereas the authors just have 5. Suggestion to replace with 'in Human visual cortex' or 'in object-selective and category-selective regions in human visual cortex' (see related next point).

(2) Given that the normalization model is a favorite in the attended conditions but less so in the other conditions, the abstract and title could be toned down, as this gives the impression that the normalization model consistently outperforms the other models.

(3) Lines 18-20 – Include more recent references

(4) Line 21 – Explain how normalization regulates the gain of the attended stimulus in the monkey brain.

(5) The authors state at multiple points in the manuscript that 'no study to date has directly tested the validity of the normalization model in predicting human cortical responses". Which is odd because the authors then go to cite a few of the studies that have done exactly that. Moreover, there are a number of other studies left out of, such as Ithipuripat et al., 2014. Perhaps the authors mean 'using fMRI', but in that case there still remains a few (including that cited) that suggest otherwise. The authors should soften this statement, or clarify on what they mean.

(6) Lines 23-25 – Related to the previous point. There is a recent study in PNAS which must be cited Aqil, Knapen and Dumoulin (PNAS 2021). Moreover, the authors should discuss how this manuscript goes beyond that study and discuss any supporting or conflicting conclusions. See also the Commentary by Foster and Ling (PNAS 2021).

Also, there is evidence of normalization in behavioral studies that seems worth mentioning; e.g., Herrmann et al. (Nature Neurosci 2010, JoV 2012); Schwedhelm et al. (PLoS Comput Biol 2016).

(7) The authors should discuss if there is neural and/or behavioral evidence of normalization in object-based attention (which would also provide some justification for the stimuli chosen, rather than what prior work of a similar vein has used i.e. contrast gratings). This is important as it is the manipulation in the present study, but most of the papers in the Introduction refer to spatial attention, and it should not be taken for granted that either the behavioral or the neural effects are the same for different types of attention or for different types of stimuli.

(8) Line 33: Specify which models? The normalization and weighted average model?

(9) Line 55-56. The authors refer to the figure to show how BOLD response changes across task conditions, but there is no interpretation of these data. How do we expect the raw data to look? Does this match the current data? If so why, if not, why not? In its current form, a figure of data is shown without explaining what it means. The authors could present examples of the BOLD signal elicited from the seven conditions; doing so would provide a deeper look at the data and how the fMRI signal changes with condition. Is there a pattern to the response?

(10) Line 86-89: The authors state, "As evident in the figure, the predictions of the normalization model (in orange) very closely follow the observed data (in navy), while the predictions of the weighted sum and weighted average models (light blue and gray, respectively) have significant deviations from the data". In which ROIs? All of them? It looks like the normalization model looks close to the data for PFs, EBA and LO. For V1 and PPA it is better than the other two but it is not as close. Is this what is shown in 2C?

Suggestion to order the panels in Figure 2 logically (the BOLD and model data all presented in A for all 5 ROIs) – there's no reason to just show LO1 as the main figure because it looks best, then the variance explained panel after – then separately the R2. Much of the paper comes across as 'normalization is better for all' and removes any nuance in differences in different areas; why might it be better in some than others? This is not detailed in the manuscript.

(11) Could the asymmetry effect found in the BOLD response be driven potentially by nonlinearities in the BOLD response, rather than being neural in origin? Do the authors have ways to mitigate this concern? One thought would be to show the relation (or lack thereof) between voxels that have a high BOLD response, and their degree of asymmetry. If the effect is not due to saturation, one would expect no significant relation.

(12) The changes in response with attention and the asymmetry effects are predictions of the biased competition model of attention and reference should be made to this literature (e.g., papers by Desimone and Duncan). In addition, this is not the first time that an asymmetry effect is observed in the fMRI literature (line 131). This effect is a direct prediction of the biased competition theory of attention, and has been previously reported in the fMRI literature as well (e.g., in reference 12).

(13) Line 178-181 – The limitation mentioned here is very vague. Some discussion of the limitations should go beyond 'we are aware there are limitations and hope one day they can be overcome'. What do these limitations mean for the data analysis and the interpretation of results? What could alternative explanations be? How will they be overcome?

Model

(1) A main concern regarding the interpretation of these results has to do with the sparseness of data available to fit with the models. The authors pit two linear models against a nonlinear (normalization) model. The predictions for weighted average and summed models are both linear models doomed to poorly match the fMRI data, particularly in contrast to the nonlinear model. So, while the verification that responses to multiple stimuli don't add up or average each other is appreciated, the model comparisons seem less interesting in this light. The model testing endeavor seems rather unconstrained. A 'true' test of the model would likely need a whole range of contrasts tested for one (or both) of the stimuli. Otherwise, as it stands we simply have a parameter (σ) that instantly gives more wiggle room than the other models. It would be fairer to pit this normalization model against other nonlinear models. Indeed, this has already been done in previous work by Kendrick Kay, Jon Winawer and Serge Dumoulin's groups. The same issue of course extends to the attended conditions.

(2) The difference in the number of free parameters biases the results. The normalization model has more free parameters than the other two models. The authors use a split-half approach to avoid the problem of overfitting, but a model with more free parameters in and of itself potentially has a better capacity to fit the data compared to the other models. This built in bias could account for the results. Model comparison (like with the AIC measure) is necessary to account for this difference.

(3) Related to the above point, the response to the Pref (P) and Null (N) stimuli are also free parameters in the normalization model and it's not clear why the values of Rp and Rn are not used here as for the other two models? These differences could again account for the better fit observed for the normalization model. Further, what are the values of Cp and Cn in the normalization model? How are they determined?

(4) The weighted average model is confusing. Equation 5a is a strict average model. The weighted average model (Desimone and Duncan) however proposes that the response to combined stimuli is a weighted average of the response to the individual stimuli (i.e., with possibly different weights for each stimulus). Second, attention biases these weight values by possibly different amounts for different stimuli/in different brain regions. In this study, the response to the combined stimuli is modeled as a strict average (with a fixed weight of 0.5), and further, one fixed weight (β) is assigned to P and N for attention effects. However, in the weighted average model, the effect of attention could potentially be different for the preferred and null stimuli (i.e., a different β for P and another β for N), which might generate a better fit for the weighted average model. Indeed, although it is difficult to read off the graph, attention appears to enhance the response to the N stimuli more than to the P stimuli. How might the results be affected if the weights in the PN condition are not fixed to 0.5, and if attention is allowed to differently affect the two types of stimuli? A similar argument could also be made for the weighted sum model.

(5) Line 57-61: What is the justification of using the weighted sum model when we know that this model is not reflective of what is going on in the brain? It sounds like it has been set up to fail. Has the weighted sum model been shown to predict responses in prior work? It is never really brought up again in the discussion, so its purpose is unclear. Wouldn't a comparison of the weighted average and normalization model make more sense? Better yet, consider comparisons with other model(s) beyond normalization that could explain the data.

Methods and data analysis

(1) FIGURE 1 – protocol – Is task difficulty equated across house category, body category, and color change at fixation?

(2) Did participants complete any practice trials for the one back task, or just went right into the scanner? Was the accuracy of the participants in the one back task recorded to ensure that they were paying attention and completing the task properly? Same for the fixation task

(3) Most of page 6 is a repetition of the methods (much copy and paste), not really needed, instead a short summary would be better or just move the methods to after the introduction.

(4) Some of the details regarding the actual design of the fMRI experiment seem glossed over. From what I can gather, each block was only a few seconds, with no rest between blocks? If so, the BOLD response would be exceptionally driven to saturating levels throughout the experiment. That said, it is difficult to unpack the details based on the Methods section as vague detail is provided.

(5) Lines 47-51. It is unclear how preferred (P) or null (N) stimulus categories were assigned to each voxel. Does a voxel preferring houses respond greater to houses than a body? Was there a statistical measure for determining what was the P or N stimulus for a given voxel? We imagine that almost all voxels in EBA significantly preferred bodies, and almost all voxels in PPA preferred houses. If so, how is this accounted for? Was there a reliable, statistically significant difference in house or body preference for voxels in the other ROIs, or is the difference between response activation to the two categories marginal? How many voxels preferred houses vs. bodies in each ROI? For each voxel, how reliable is the attribution of the P or N stimulus (e.g., did you use a cross-validation approach and determine P/N on some runs and tested the reliability on the remaining runs)?

(6) Were the P and N categories determined on all the data (a risk of double-dipping) or only on one half of the dataset?

(7) Provide more details of the GLM. What were the regressors?

(8) Lines 79-81: Were the model parameters determined in the second half and predictions made in the first half as well? Would averaging the two directions provide more robust results?

(9) Category stimuli were fit within a 10.2 degree square of visual angle. The methods describe the V1 localizer as a wedge 60 degree in angle. What was the stimulus extent (i.e. eccentricity) of the wedge? Was it also 10.2 degrees so that the V1 ROI encompassed the same amount of visual space as the category stimuli? Same question for the Category localizer. Were these stimuli also 10.2 degrees? Were the ROIs (at least for V1?) defined within the same eccentricity range so that the ROI analysis was confined to where the category stimuli were presented in the visual field?

(10) What software was used to define LOC PF EBA and PPA? And what metric do we have to ensure that these are reliable ROIs? The authors would benefit by including figures that show examples of the ROIs overlaid on contrast data for the localizer conditions. The same goes for the β weights within the ROIs.

(11) Preprocessing – how were the data aligned?

(12) The paper refers to a Supplementary Methods, which I could not find.

Results and statistics

(1) What was accuracy like on these tasks? There was no description of performance at all. It is necessary to provide this information to assess the attentional manipulation

(2) Line 96 – What is ts? T-statistic? Did you run a series of t-tests comparing normalization vs noise ceiling? Why not an ANOVA and correct for multiple comparisons? And what does it mean that the R2 was not significant different from noise ceiling? Interpretation of these results must be provided.

(3) Lines 119-123. Again, why a t-test statistic here? Did you run multiple t-tests for each ROI? Shouldn't this be an ANOVA across the 5 ROIs? As it stands, line 120 says that you ran multiple t tests across all the regions, and then report single t-statistic? The paper needs to be clear (and correct) in the type of statistics it uses to make its claims.

(4) There is no assessment of the quality of the data – no Β weight maps showing the responses on the brain, or what proportion of voxels in each ROI actually responded to the stimuli. A lot of interesting data that could be shown is reduced to line plots -this comes across as the paper starting at the end, rather than providing any sense of buildup in the data analysis and the conclusions that the authors draw.

(5) The authors show the goodness of fit across the 5 modeled conditions in 2C – fine, but what is the justification for lumping all the conditions together? Is there a supplement figure that teases this apart, with the goodness of fit for each model for each of the conditions, for each ROI? It would help to be comprehensive.

(6) Averaging all the data together (e.g., in Figure 2c across all conditions) does not provide a clear picture of the results. Is it possible that it is only in the attention conditions that the normalization model outperforms the others (although again, note the remark about the different numbers of free parameters in each model)? It would be more transparent to plot Figure 2c separately for the isolated/paired conditions, even though the story might become more complex.

(7) Related to the above points, the data are provided in a very synthesized manner, and raw results are not shown. For example, could we see the values of the estimated model parameters? How do they compare when determined in each half of the data (to get a sense of reliability)?

Figures

(1) Figure 1b: The gray squares overlap a lot so that we don't see the stimuli in each gray square.

(2) The y-axis label in Figure 2A is not BOLD response (this would be appropriate if the figure was a BOLD time series); is this the average fMRI Β weight?

(3) In Figure 2 (a and b), the connecting of data points as if they are on a continuum is not correct given their categorical nature. Moreover, the figure could be improved so that the actual data points are made more prominent, and the model predictions are dashed lines. What exactly do the error bars in Figure 2a and 2b correspond to?

(4) Suggestion to order the panels in Figure 2 logically (the BOLD and model data all presented in A for all 5 ROIs).

(5) Is there any statistic to help evaluate the data in Figure 2C? (Line 93)

(6) Figure 3 – Why not include the BOLD response for P only and N only? That would be informative.

---

## [Author Response]

Essential Revisions:Theoretical framework and interpretation(1) The title is misleading and a bit grandiose – "In" instead of "across". Across would imply evidence for normalization across all ~30 areas of visual brain, whereas the authors just have 5. Suggestion to replace with 'in Human visual cortex' or 'in object-selective and category-selective regions in human visual cortex' (see related next point).

We changed the title from “Evidence for Normalization as a Fundamental Operation Across the Human Visual Cortex” to “Evidence for Normalization as a Fundamental Operation in the Human Visual Cortex”.

(2) Given that the normalization model is a favorite in the attended conditions but less so in the other conditions, the abstract and title could be toned down, as this gives the impression that the normalization model consistently outperforms the other models.

Based on comment 9 from the reviewers in the *Methods and Data Analysis* section of this letter, we changed our analyses to only include active voxels in each ROI to account for the differences in the stimulus size between the main task and the localizer tasks. This extra step improved our results so that the normalization model is now better than the weighted average model for both the attended and unattended conditions. As such, we decided to keep the title. Nevertheless, to more accurately convey the results, we have slightly modified the sentence in the abstract:

“Focusing on the primary visual area V1, the object-selective regions LO and pFs, the body-selective region EBA, and the scene-selective region PPA, we first modeled single-voxel responses using a weighted sum, a weighted average, and a normalization model and demonstrated that although the weighted sum and the weighted average models also made acceptable predictions in some conditions, the response to multiple stimuli could generally be better described by a model that takes normalization into account.”

(3) Lines 18-20 – Include more recent references

We have now included more recent references in the Introduction section in lines 50-54:

“In addition to regional computations for multiple-stimulus representation, the visual cortex relies on top-down mechanisms such as attention to select the most relevant stimulus for detailed processing (Moran and Desimone 1985, Desimone and Duncan 1995, Chun et al. 2011, Baluch and Itti 2011, Noudoost et al. 2010, Maunsell 2015, Thiele and Bellgrove 2018, Itthipuripat et al. 2014, Moore et al. 2017, Buschman and Kastner 2015).”

(4) Line 21 – Explain how normalization regulates the gain of the attended stimulus in the monkey brain.

We have replaced the sentence with a short description in the Introduction section in lines 56-60:

“Previous studies have demonstrated how the normalization computation accounts for these observed effects of attention in the monkey brain. They have suggested that normalization attenuates the neural response in proportion to the activity of the neighboring neuronal pool (Reynolds Heeger 2009, Lee Maunsell 2009, Boynton 2009, Ni et al. 2012).”

(5) The authors state at multiple points in the manuscript that 'no study to date has directly tested the validity of the normalization model in predicting human cortical responses". Which is odd because the authors then go to cite a few of the studies that have done exactly that. Moreover, there are a number of other studies left out of, such as Ithipuripat et al., 2014. Perhaps the authors mean 'using fMRI', but in that case there still remains a few (including that cited) that suggest otherwise. The authors should soften this statement, or clarify on what they mean.

We thank the reviewer for pointing out the apparent paradox in our phrasing. We have modified the introduction and hope that the following paragraph can better convey the contribution of our study (added in the Introduction section in lines 73-82):

“Although previous studies have qualitatively suggested the role of normalization in the human visual cortex (Kliger and Yovel 2020, Itthipuripat et al. 2014, bloem and Ling 2019), evidence for directly testing the validity of the normalization model in predicting human cortical responses in a quantitative way remains scarce. A few studies have demonstrated the quantitative advantage of normalization-based models compared to linear models in predicting human fMRI responses using gratings, noise patterns, and single objects (Kay et al. 2013a, Kay et al. 2013b), as well as moving checkerboards (Aqil et al. 2021, Foster and Ling 2021). However, whether normalization can also be used to predict cortical responses to multiple objects, and if and to what extent it can explain the modulations in response caused by attention to objects in the human brain remain unanswered.”

(6) Lines 23-25 – Related to the previous point. There is a recent study in PNAS which must be cited Aqil, Knapen and Dumoulin (PNAS 2021). Moreover, the authors should discuss how this manuscript goes beyond that study and discuss any supporting or conflicting conclusions. See also the Commentary by Foster and Ling (PNAS 2021).Also, there is evidence of normalization in behavioral studies that seems worth mentioning; e.g., Herrmann et al. (Nature Neurosci 2010, JoV 2012); Schwedhelm et al. (PLoS Comput Biol 2016).

We have added citations of the study by Aqil et al., and the commentary by Foster and Ling, as well as how our study goes beyond that study in the Introduction section in lines 76-82:

“A few studies have demonstrated the quantitative advantage of normalization-based models compared to linear models in predicting human fMRI responses using gratings, noise patterns, and single objects (Kay et al. 2013a, Kay et al. 2013b), as well as moving checkerboards (Aqil et al. 2021, Foster and Ling 2021). However, whether normalization can also be used to predict cortical responses to multiple objects, and if and to what extent it can explain the modulations in response caused by attention to objects in the human brain remain unanswered.”

We have also added citations of behavioral studies that report evidence of normalization in the Introduction section in lines 69-72:

“In the human visual cortex, normalization has been speculated to underlie response modulations in the presence of attention, with evidence provided both by behavioral studies of space-based (Hermann et al. 2010) and feature-based (Hermann et al. 2012, Schweldhelm et al. 2016) attention as well as neuroimaging studies of feature-based attention (Bloem and Ling 2019).”

(7) The authors should discuss if there is neural and/or behavioral evidence of normalization in object-based attention (which would also provide some justification for the stimuli chosen, rather than what prior work of a similar vein has used i.e. contrast gratings). This is important as it is the manipulation in the present study, but most of the papers in the Introduction refer to spatial attention, and it should not be taken for granted that either the behavioral or the neural effects are the same for different types of attention or for different types of stimuli.

We agree with the reviewer that the effects of space-based attention cannot be readily generalized to object-based attention. As the reviewer has pointed out, this is precisely why we ran this study. We could not find evidence of normalization in object-based attention in the literature. However, there is evidence for normalization in feature-based attention. We have now separated the references related to feature- and space-based attention and have made our contribution related to object-based attention more explicit in the Introduction section in lines 56-68:

“Previous studies have demonstrated how the normalization computation accounts for these observed effects of attention in the monkey brain. They have suggested that normalization attenuates the neural response in proportion to the activity of the neighboring neuronal pool (Reynolds Heeger 2009, Lee Maunsell 2009, Boynton 2009, Ni et al. 2012). These studies have focused on space-based (Reynolds Heeger 2009, Lee Maunsell 2009, Ni et al. 2012,) or feature-based (Ni et al. 2019) attention. While it has been suggested that these different forms of attention affect neural responses in similar ways, there exist distinctions in their reported effects, such as different time courses (Hayden and Gallant 2005), and the extent to which they affect different locations in the visual field (Serences and Boynton 2007, Womelsdorf et al. 2006), suggesting that there are common sources as well as differences in modulation mechanisms between these forms of attention (Ni and Maunsell 2019). This leaves open the question of whether normalization can explain the effects of object-based attention.”

(8) Line 33: Specify which models? The normalization and weighted average model?

We have included the name of the models in the sentence to avoid confusion in the Introduction section in lines 89-92:

“We also demonstrate that normalization is closer to averaging in the absence of attention, as previously reported by several studies, but that the results of the weighted average model and the normalization model diverge to a greater extent in the presence of attention.”

(9) Line 55-56. The authors refer to the figure to show how BOLD response changes across task conditions, but there is no interpretation of these data. How do we expect the raw data to look? Does this match the current data? If so why, if not, why not? In its current form, a figure of data is shown without explaining what it means. The authors could present examples of the BOLD signal elicited from the seven conditions; doing so would provide a deeper look at the data and how the fMRI signal changes with condition. Is there a pattern to the response?

We thank the reviewer for this suggestion. We agree that including the response in the seven conditions would be very helpful for readers. We have added a figure illustrating the average fMRI responses across voxels for all conditions and in all regions, see Figure 2.

We have also included our interpretation of the observed results in the *Results section* in lines 118-150:

“To examine the cortical response in different task conditions, we fit a general linear model and estimated the regression coefficients for each voxel in each condition. Figure 2a-e illustrates the average voxel coefficients for different conditions in the five regions of interest (ROIs), including V1, LO, pFs, EBA, and PPA. Each task condition was named based on the presented stimuli and the target of attention, with B and H denoting the presence of body and house stimuli, respectively, and the superscript “at” denoting the target of attention. Note that we have not included the responses related to the fixation block with no stimulus since this condition was only used to select the voxels that were responsive to the presented stimuli in each ROI (see Methods). We observed that the average voxel coefficients related to the four conditions in which attention was directed to the body or the house stimuli (the first four conditions, B^at^, B^at^H, BH^at^, H^at^) were generally higher than the response related to the last three conditions (B, H and BH conditions) in which the body and house stimuli were unattended (ts > 4, ps <1.7⨯10^-3^, corrected). This is in agreement with previous research indicating that attention to objects increases their cortical responses (Roelfsema 1998, O’craven 1999, Reddy 2009).

Looking more closely at the results in the regions EBA and PPA that have strong preferences for body and house stimuli, respectively, it seems that the effect of attention interacts with the regions’ preference. For instance, in the body-selective region EBA, the response to attended body stimuli in isolation is similar to the response to attended body stimuli paired with unattended house stimuli (compare B^at^ and B^at^H bars). On the other hand, the response to attended house stimuli in the isolated condition is significantly less than the response to attended house stimuli paired with unattended body stimuli. We can observe similar results in PPA, but not in V1 or the object-selective regions LO and pFs. But note that the latter three regions do not have strong preferences for one of the stimuli. Therefore, in order to examine the interaction between attention and preference more closely, we determined preferences at the voxel level in all ROIs.

We defined the preferred (P) and null (N) stimulus categories for each voxel in each ROI according to the voxel’s response to isolated body and isolated house conditions. Figure 2f shows the percentage of voxels in each region that were selective to bodies and houses averaged across participants. As illustrated in the figure, in the object-selective regions LO and pFs, almost half of the voxels were selective to each category, while in the EBA and PPA regions, the general preference of the region prevailed (Even though these regions are selected based on their preference), the noise in the fMRI data and other variations due to imperfect registration could lead to some voxels showing different preferences in the main session compared to the localizer session (Peelen and Downing 2005).”

In addition, we have now included the interpretation of the results illustrated in figure 3 (figure 2 in the previous version) in lines 160-173:

“We observed that the mean voxel response was generally higher when each stimulus was attended compared to the condition in which it was ignored. For instance, the response in the P^at^ condition (in which the isolated preferred stimulus was attended) was higher than in the P condition (where the isolated preferred stimulus was ignored) in LO, pFs, and PPA (ts > 3.6, ps < 0.01, corrected), marginally higher in EBA (t(18)=2.69, p=0.072, corrected), and not significantly higher in V1 (t(18)=2.52, p=0.1, corrected). Similarly, comparing the N and N^at^ conditions in each ROI, we observed an increase in response caused by attention in all ROIs (ts > 4, ps < 3.7⨯10^-3^, corrected) except for V1 (t(18)=2.4, p=0.13, corrected). A similar trend of response enhancement due to attention could also be observed in the paired conditions: attending to either stimulus increased the response in all ROIs (ts > 4.4, ps < 1.5⨯10^-3^, corrected) except for V1 (ts<2.59, ps > 0.087, corrected). In all cases, the effect of attention was absent or only marginally significant in V1, which is not surprising since attentional effects are much weaker (Mcadams and Maunsell 1999) or even absent (Luck et al. 1997) in V1 compared to the higher-level regions of the occipito-temporal cortex.”

(10) Line 86-89: The authors state, "As evident in the figure, the predictions of the normalization model (in orange) very closely follow the observed data (in navy), while the predictions of the weighted sum and weighted average models (light blue and gray, respectively) have significant deviations from the data". In which ROIs? All of them? It looks like the normalization model looks close to the data for PFs, EBA and LO. For V1 and PPA it is better than the other two but it is not as close. Is this what is shown in 2C?Suggestion to order the panels in Figure 2 logically (the BOLD and model data all presented in A for all 5 ROIs) – there's no reason to just show LO1 as the main figure because it looks best, then the variance explained panel after – then separately the R2. Much of the paper comes across as 'normalization is better for all' and removes any nuance in differences in different areas; why might it be better in some than others? This is not detailed in the manuscript.

We thank the reviewer for this suggestion. We agree that a logical order of the panels is more appropriate. We have now ordered the panels more logically (V1- Lo- pFs- EBA- PPA), all with the same size (see Figure 3):

To answer the reviewer’s question about the differences across areas, after we selected active voxels in each region for further analyses (this was done to answer comment 9 from the reviewers in the *Methods and Data Analysis* section of this letter), we calculated the normalization model’s r-squared difference from the noise ceiling. This measure was not significantly different across ROIs (repeated measures ANOVA, *F*(4,72)=0.58, *p* = 0.61). Therefore, we can say that the fit of the normalization model is not significantly worse in any of the ROIs. However, we have toned down the point about the superiority of the normalization model since its predictions are not similarly close to the data across all conditions in the *Results section* lines 218-220:

“As evident in the figure, the predictions of the normalization model (in orange) are generally better than the predictions of the weighted sum and the weighted average models (light blue and gray, respectively) in all regions. “

We have also included the results of the ANOVA test comparing the difference between the noise ceiling and the normalization fit across ROIs, showing that there is no significant difference in this measure across ROIs in the *Results section* in lines 240-246:

“We then calculated the normalization model’s r-squared difference from the noise ceiling (NRD) for each ROI. NRD is a measure of the ability of the model in accounting for the explainable variation in the data; the lower the difference between the noise ceiling and a model’s goodness of fit, the more successful that model is in predicting the data. We ran a one-way ANOVA to test for the effect of ROI on NRD, and observed that this measure was not significantly different across ROIs (F(4,72)=0.58, p = 0.61), demonstrating that the normalization model was equally successful across ROIs in predicting the explainable variation in the data.”

We have also included the details of model comparison, showing which model performed better in each region, in the *Results section* in lines 229-239:

“We first compared the goodness of fit of the three models across the five ROIs using a 3⨯5 repeated measures ANOVA. The results showed a significant main effect of model (F(2,36) = 72.9, p = 9.86⨯10^-11^) and ROI (F(4,72) = 26.66, p = 1.04⨯10^-7^), and a significant model by ROI interaction (F(8,144) = 24.96, p = 3.74⨯10^-15^). On closer inspection, the normalization model was a better fit to the data than both the weighted sum (ps < 0.019, corrected) and the weighted average (ps < 5.7⨯10^-5^, corrected) models in all ROIs. Since the normalization model had more parameters, we also used the AIC measure to correct for the difference in the number of parameters. The normalization model was a better fit according to the AIC measure as well (see supplementary file 2).

It is noteworthy that while the weighted average model performed better than the weighted sum model in LO and EBA (ps < 0.0016, corrected), it was not significantly better in pFs and PPA (ps > 0.37, corrected), and worse than weighted sum in V1 (p = 6.5⨯10^-6^, corrected).”

We have discussed the difference in the fit of the weighted average and weighted sum models across regions in the *Discussion section* in lines 422-430:

“Our results demonstrate that while the weighted average model generally performs better than the weighted sum model in the higher-level occipito-temporal cortex, the weighted sum model provides better predictions in V1. These results suggest stronger sublinearity in higher regions of the visual cortex compared to V1, which is in agreement with previous reports (Kay et al. 2013b). This observation might be related to the higher sensitivity of V1 neurons to contrast (Goodyear and Menon, 1998), causing a more significant decrease in V1 responses to low-contrast stimuli. This, in turn, might make the low-contrast stimulus weaker for V1 neurons, causing a move towards a lower level of sublinearity (Sceniak et al. 1999).”

(11) Could the asymmetry effect found in the BOLD response be driven potentially by nonlinearities in the BOLD response, rather than being neural in origin? Do the authors have ways to mitigate this concern? One thought would be to show the relation (or lack thereof) between voxels that have a high BOLD response, and their degree of asymmetry. If the effect is not due to saturation, one would expect no significant relation.

The reviewer raises a very good point.

To examine whether the observed asymmetry effect was driven by BOLD response saturation, we tested a saturation model (the weighted average UWUB saturation, detailed in the *Methods* section). This model’s goodness of fit, as well as its predictions of response change and asymmetry, are illustrated in the figure 5b,c in the paper:

As illustrated in the figure 5 panel c, the saturation model’s predicted asymmetry was closer to the data than normalization’s prediction only in EBA (*p* = 0.043, corrected). In other regions, the normalization model’s prediction of asymmetry was either significantly closer to the data (in V1, LO and pFs, *ps* < 1.02⨯10^-6^ , corrected) or not significantly different from the saturation model (in PPA *p* = 0.63, corrected).

We then compared the overall fits of the two models by running a 2⨯5 ANOVA across ROIs. We observed a significant effect of model (*F*(1,18) = 91.16, *p* = 1.8⨯10^-8^), a significant effect of ROI (*F*(4,72) = 19.46, *p* = 2.21⨯10^-7^), and a significant model by ROI interaction (*F*(4,72) = 4.82, *p* = 0.0061). Post-hoc t-tests showed that the normalization model was a significantly better fit than the saturation model in all ROIs (ps< 0.008, corrected).

Note that saturation is a characteristic of cortical responses even at the neural level. Our results are in agreement with previous monkey electrophysiology studies, showing the existence of an asymmetry in attentional modulation for attention to the preferred versus the null stimulus and demonstrating that the normalization model can explain this effect (Ni and Maunsell, 2012). Since the asymmetry in attentional modulation has also been previously reported at the neural level (Ni and Maunsell, 2012), this effect is unlikely to be only related to the nonlinearities in the BOLD signal.

We have added these results in the *Results section* in lines 345-357:

“Next, to explore whether the observed asymmetry was caused by response saturation, we tested a nonlinear variant of the weighted average model with saturation (the weighted average UWUB saturation model). This model’s goodness of fit, as well as its predictions of response change and asymmetry, are illustrated in figure 5b,c. As illustrated in the figure, the saturation model’s predicted asymmetry was closer to the data than normalization’s prediction only in EBA (p = 0.043, corrected). In other regions, the normalization model’s prediction of asymmetry was either significantly closer to the data (in V1, LO and pFs, ps < 1.02⨯10^-6^ , corrected) or not significantly different from the saturation model (in PPA, p = 0.63, corrected).

After running a 2⨯5 ANOVA to compare the fits of the normalization model and the weighted average UWUB saturation model across ROIs, we observed a significant effect of model (F(1,18) = 91.16, p = 1.8⨯10^-8^), a significant effect of ROI (F(4,72) = 19.46, p = 2.21⨯10^-7^), and a significant model by ROI interaction (F(4,72) = 4.82, p = 0.0061). Post-hoc t-tests showed that the normalization model was a significantly better fit than the saturation model in all ROIs (ps< 0.008, corrected).”

We have also included discussions on asymmetry in the *Discussion section* in lines 468-479:

“Another limitation in interpreting the results is related to a possible stronger saturation of the BOLD response, which can potentially explain the observed asymmetry in attentional modulation. Since the asymmetry in attentional modulation has also been previously reported at the neural level (Ni and Maunsell, 2012), this effect is unlikely to be exclusively caused by the nonlinearities in the BOLD signal. It is noteworthy, however, that saturation is a characteristic of cortical responses even at the neural level. Whether this effect at the neural level is caused by response saturation or as a result of the normalization computation cannot be distinguished from our current knowledge. Nevertheless, we tested a variant of the weighted average model with an extra saturation parameter. Although this model could partially predict the observed asymmetry, the predictions were worse than the normalization model’s predictions. Also, this model was an overall worse fit to the data compared to the normalization model. The normalization model, therefore, provides a more parsimonious account of the data.”

(12) The changes in response with attention and the asymmetry effects are predictions of the biased competition model of attention and reference should be made to this literature (e.g., papers by Desimone and Duncan). In addition, this is not the first time that an asymmetry effect is observed in the fMRI literature (line 131). This effect is a direct prediction of the biased competition theory of attention, and has been previously reported in the fMRI literature as well (e.g., in reference 12).

The biased competition model predicts the change in response with attention (reduction in response when shifting attention from preferred to null stimuli), and we have added references to the related work in the *Discussion section*. However, our modeling results showed that the predictions of the weighted average model (as a quantitative derivation of the biased competition model) were significantly lower for the amount of change in response by attention than what was observed in the data. On the other hand, this response change was closely predicted by the normalization model.

We have included reference to the biased competition model in the *Discussion section* in lines 377-382:

“Although this response reduction has also been predicted by the biased competition model (Desimone and Duncan 1995), our results showed that the predictions of the weighted average model were significantly lower than the response reduction observed in the data in all ROIs except in V1. In contrast, the normalization model predicted the response reduction more accurately in all regions except in V1, where no significant response reduction was observed.”

Regarding the asymmetry effect, we are unaware of any previous studies showing or predicting the asymmetry effect in the framework of the biased competition model. In fact, Reddy et al. (2009) used a weighted average model and reported a bias of 30% when attention was directed to one stimulus compared to when it was divided between the two stimuli. They did not sort conditions based on voxel preference for the presented stimuli and reported an overall 30% weight shift with attention, with no significant main effect of category, which suggests no asymmetry.

According to our modeling results, the weighted average variant similar to the one used by Reddy et al. (the “weighted average UW” model in the *Other variants of the weighted average model* section) did not predict such an asymmetry. A nonlinear variant of the weighted average model in which we added a saturation parameter to the linear weighted average version could predict this effect, but the predictions were not as close to the data as the normalization model.

(13) Line 178-181 – The limitation mentioned here is very vague. Some discussion of the limitations should go beyond 'we are aware there are limitations and hope one day they can be overcome'. What do these limitations mean for the data analysis and the interpretation of results? What could alternative explanations be? How will they be overcome?

We have added the following sentences to the *Discussion section* in lines 455-479 to elaborate on the limitations of our study:

“It is noteworthy that here, we are looking at the BOLD responses. We are aware of the limitations of the fMRI technique as the BOLD response is an indirect measure of the activity of neural populations. While an increase in the BOLD signal could be related to an increase in the neuronal firing rates of the local population (Logothetis et al. 2001), it could also be related to subthreshold activity resulting from feedback from the downstream regions of the visual cortex (Heeger and Ress 2002). The observed effects, therefore, may be related to local population responses or may be influenced by feedback from downstream regions. Also, since the measured BOLD signal is related to the average activity of the local population, and we do not have access to single unit responses, some effects may change in the averaging process. Nevertheless, our simulation results show that the effects of the normalization computation are preserved even after averaging. We should keep in mind, though, that these are only simulations and are not based on data directly measured from neurons. Future experiments with intracranial recordings from neurons in the human visual cortex would be invaluable in validating our results.

Another limitation in interpreting the results is related to a possible stronger saturation of the BOLD response, which can potentially explain the observed asymmetry in attentional modulation. Since the asymmetry in attentional modulation has also been previously reported at the neural level (Ni and Maunsell, 2012), this effect is unlikely to be exclusively caused by the nonlinearities in the BOLD signal. It is noteworthy, however, that saturation is a characteristic of cortical responses even at the neural level. Whether this effect at the neural level is caused by response saturation or as a result of the normalization computation cannot be distinguished from our current knowledge. Nevertheless, we tested a variant of the weighted average model with an extra saturation parameter. Although this model could partially predict the observed asymmetry, the predictions were worse than the normalization model’s predictions. Also, this model was an overall worse fit to the data compared to the normalization model. The normalization model, therefore, provides a more parsimonious account of the data.”

Model(1) A main concern regarding the interpretation of these results has to do with the sparseness of data available to fit with the models. The authors pit two linear models against a nonlinear (normalization) model. The predictions for weighted average and summed models are both linear models doomed to poorly match the fMRI data, particularly in contrast to the nonlinear model. So, while the verification that responses to multiple stimuli don't add up or average each other is appreciated, the model comparisons seem less interesting in this light. The model testing endeavor seems rather unconstrained. A 'true' test of the model would likely need a whole range of contrasts tested for one (or both) of the stimuli. Otherwise, as it stands we simply have a parameter (σ) that instantly gives more wiggle room than the other models. It would be fairer to pit this normalization model against other nonlinear models. Indeed, this has already been done in previous work by Kendrick Kay, Jon Winawer and Serge Dumoulin's groups. The same issue of course extends to the attended conditions.

We thank the reviewer for this comment. Regarding the reviewer’s concern about comparing a nonlinear model with a linear one, we have tried to answer with two different approaches. First, as suggested by the reviewer, we used a nonlinear variant of the weighted average model with an extra saturation parameter. This model had distinct free parameters for the weights of the preferred and null stimuli and distinct attention parameters for the preferred and null stimuli. It, therefore, had five free parameters. The comparison between the results of this model with the normalization model’s predictions showed that the predictions of this model were significantly worse than normalization in all regions (*ps* < 0.008, corrected).

We have added these results in the *Results section* in lines 352-357:

“After running a 2⨯5 ANOVA to compare the fits of the normalization model and the weighted average UWUB saturation model across ROIs, we observed a significant effect of model (F(1,18) = 91.16, p = 1.8⨯10^-8^), a significant effect of ROI (F(4,72) = 19.46, p = 2.21⨯10^-7^), and a significant model by ROI interaction (F(4,72) = 4.82, p = 0.0061). Post-hoc t-tests showed that the normalization model was a significantly better fit than the saturation model in all ROIs (ps< 0.008, corrected).”

We have also included a figure illustrating the comparison between the normalization model and the saturation model in the paper (see the figure in the response to comment 11 of *Theoretical framework and interpretation* section of this letter).

The second approach we took was the simulation of responses of a neural population. We simulated three neural populations: one in which the neural response to multiple stimuli and attended stimuli was calculated based on the weighted sum model (summing neurons), one in which neural responses were calculated based on the weighted average model (averaging neurons), and one in which neural responses were calculated based on divisive normalization (normalizing neurons). Then, by randomly selecting neurons from the population to make up voxels, we estimated model parameters for each voxel using each of the three models used in our study. We then predicted the response of the left-out half of the simulated data using the parameters estimated for the first half. The results demonstrate that the normalization model is only a better fit when predicting the response of a normalizing population (figure 3—figure supplement 2). It performs worse than the weighted sum model for summing neurons and worse than the weighted average model for averaging neurons. Therefore, a nonlinear model is not necessarily a better fit for a linear operation.

We have included the simulation results in the *Results section* in lines 258-267:

“To ensure that the superiority of the normalization model over the weighted sum and weighted average models was not caused by the normalization model's nonlinearity or its higher number of parameters, we ran simulations of three neural populations. Neurons in each population calculated responses to multiple stimuli and attended stimuli by a summing, an averaging, and a normalizing rule (see Methods). We then used the three models to predict the population responses. Our simulation results demonstrate that despite the higher number of parameters, the normalization model is only a better fit for the population of normalizing neurons and not for summing or averaging neurons, as illustrated in figure 3—figure supplement 2. These results confirm that the better fits of the normalization model cannot be related to the model's nonlinearity or its higher number of parameters.”

We have also included the simulation details in the *Simulation* section in the *Methods* section.

In addition to these two approaches, and along with cross-validation previously used, we have also included the AIC measure for model comparison to correct for the difference in the number of parameters in the *Results section* in lines 234-236 (see the response to the next comment):

“Since the normalization model had more parameters, we also used the AIC measure to correct for the difference in the number of parameters. The normalization model was a better fit according to the AIC measure as well (see supplementary file 2).”

Therefore, we believe that our simulation approach and the nonlinear model we tested provide enough evidence for our conclusion; that the success of the normalization model is not due to its higher number of parameters, but rather as a result of it being a closer estimation of the computation performed at the neural level for object representation. Using AIC to compensate for the difference in the number of parameters confirms our cross-validated r-squared comparisons. Given the robustness of the agreement in the results of these analyses, we do not believe nonlinearity or the number of parameters can explain the superiority of the normalization model.

We have discussed these approaches in the *Discussion section* in lines 442-454:

“Here, we compared the nonlinear normalization model with two linear models with fewer free parameters. To ensure that the difference in the number of free parameters did not affect the results, we used cross-validation and the AIC measure to compare model predictions with the data. If the success of the normalization model was due to its higher number of free parameters, it would affect its predictions for left-out data. We observed that the normalization model was also successful in predicting the left-out part of the data. In addition, we tested a nonlinear model variant with five free parameters. This model was still a worse fit than the normalization model. Finally, we used simulations of three different neural populations, with neurons in each population following either a summing, averaging, or normalization rule in their response to multiple stimuli and attended stimuli. Simulation results demonstrated that the normalization model was a better fit only for the normalizing population, confirming that the success of the normalization model is not due to its nonlinearity or the higher number of parameters but rather as a result of it being a closer estimation of the computation performed at the neural level for object representation.”

Regarding the reviewer’s suggestion on testing a range of contrasts, as we have now mentioned in the Discussion section, here, we have explored the effects of attention and whether the normalization model could explain the effects of object-based attention in the human visual cortex, which has not been previously studied. While we acknowledge the importance of studying the effects of contrast, the presentation of superimposed stimuli limits the range of contrasts we can use because high-contrast stimuli block each other, and stimuli with very low contrasts are very difficult to recognize. Besides, including variations of both attentional state and contrast would need a multi-session experiment, which was not feasible given the limitations caused by the COVID pandemic.

We have added this in the *Discussion section* in lines 431-441:

“Attention to a stimulus has been suggested in the literature to be similar to an increase in the contrast of the attended stimulus (Ni and Maunsell 2012), which is manifested in the similar effects of attention and contrast in the normalization equation. In this study, we presented the stimuli with a constant contrast but changed the number of stimuli and their attentional state to determine whether the normalization model could explain the effects of object-based attention in the human visual cortex, which has not been previously studied. Although we acknowledge that testing for a range of contrasts would help in exploring the interaction of contrast and attention and in generalizing the conclusions of this study, it would not be trivial for the current design because the presentation of superimposed stimuli limits the range of contrasts that can be tested. Besides, including variations of both attentional state and contrast would need a multi-session fMRI experiment that would limit the feasibility of the study.”

(2) The difference in the number of free parameters biases the results. The normalization model has more free parameters than the other two models. The authors use a split-half approach to avoid the problem of overfitting, but a model with more free parameters in and of itself potentially has a better capacity to fit the data compared to the other models. This built in bias could account for the results. Model comparison (like with the AIC measure) is necessary to account for this difference.

As our simulation results show (see the response to the previous comment), a non-linear model does not necessarily fit the data better if the underlying population response follows a linear summation. Our cross-validation approach ensures that the number of parameters would not benefit us in fitting the left out data (Kay et al. 2013b). Nevertheless, in addition to predicted r-squared on the left-out data, we have now used the Akaike Information Criterion (AIC) to compare models. Results are in line with our cross-validation approach. This is not surprising given the previous literature showing that the cross-validation approach leads to similar results as the AIC approach (Fang 2011, Kay et al. 2013b). The normalization model had a smaller AIC value than both the weighted sum and the weighted average models in all ROIs.

We have included the AIC comparison results between normalization and the two other models in the *Results section* in lines 232-236:

“On closer inspection, the normalization model was a better fit to the data than both the weighted sum ( ps < 0.019, corrected), and the weighted average model (ps < 5.7⨯10^-5^, corrected) models in all ROIs. Since the normalization model had more parameters, we also used the AIC measure to correct for the difference in the number of parameters. The normalization model was a better fit according to the AIC measure as well (see supplementary file 2).”

We have included the ΔAIC values for all regions in supplementary file 2.

(3) Related to the above point, the response to the Pref (P) and Null (N) stimuli are also free parameters in the normalization model and it's not clear why the values of Rp and Rn are not used here as for the other two models? These differences could again account for the better fit observed for the normalization model. Further, what are the values of Cp and Cn in the normalization model? How are they determined?

The normalization model is different in nature from the two linear models and takes into account the suppression caused by the neighboring pool of neurons, even in the presence of a single stimulus. We cannot, therefore, use the measured response in isolated conditions as the excitatory drive in paired conditions. Rather, we need extra parameters to estimate the excitation caused by each stimulus and then use this excitation to predict the response to attended and ignored stimuli in isolated and paired conditions (Ni et al. 2012, Ni and Maunsell 2017, Ni and Maunsell 2019). On the other hand, the weighted sum and the weighted average models are not concerned with the underlying excitation and suppression. The assumption of these models is based on the resulting response that we actually measure in the paired condition, considering it to be respectively the sum or the average of the measured response in the isolated conditions. In order to take the difference in the number of model parameters into account, we have used cross-validated r-squared and AIC measures. We also tested a model with five parameters (response to point 1 of the *Model* section of this letter) which was a worse fit than normalization. Moreover, our simulation results (explained in point 1 as well) also showed that a model with a higher number of parameters is not a better fit when the underlying neural computation is different from the model’s computation. Given the robustness of the agreement in the results of these analyses, we do not believe the number of parameters can explain the superiority of the normalization model.

We have added an explanation of the need for Lp and LN parameters in the *Methods* section in lines 620-631:

“It is noteworthy that the normalization model is different in nature from the two linear models and takes into account the suppression caused by the neighboring pool even in the presence of a single stimulus. We cannot, therefore, use the measured response in isolated conditions as the excitatory drive in paired conditions. Rather, we need extra parameters to estimate the excitation caused by each stimulus. We then use this excitation to predict the response to attended and ignored stimuli in isolated and paired conditions (Ni et al. 2012, Ni and Maunsell 2017, Ni and Maunsell 2019). On the other hand, the weighted sum and the weighted average models are not concerned with the underlying excitation and suppression. The assumption of these models is based on the resulting response that we actually measure in the paired condition, considering it to be respectively the sum or the average of the measured response in the isolated conditions. In order to take the difference in the number of model parameters into account, we have used both cross-validated r-squared on independent data and AIC measures (see the section on model comparison).”

As for the contrast values, we set them to one when the stimulus was presented and to zero when the stimulus was not presented, as explained in the *Methods* section in lines 615-618. We have now added a sentence for a more explicit explanation:

“c_P_ and c_N_ are the respective contrasts of the preferred and null stimuli. Zero contrast for a stimulus denotes that the stimulus is not present in the visual field. In our experiment, we set contrast values to one when a stimulus was presented and to zero when the stimulus was not presented.”

(4) The weighted average model is confusing. Equation 5a is a strict average model. The weighted average model (Desimone and Duncan) however proposes that the response to combined stimuli is a weighted average of the response to the individual stimuli (i.e., with possibly different weights for each stimulus). Second, attention biases these weight values by possibly different amounts for different stimuli/in different brain regions. In this study, the response to the combined stimuli is modeled as a strict average (with a fixed weight of 0.5), and further, one fixed weight (β) is assigned to P and N for attention effects. However, in the weighted average model, the effect of attention could potentially be different for the preferred and null stimuli (i.e., a different β for P and another β for N), which might generate a better fit for the weighted average model. Indeed, although it is difficult to read off the graph, attention appears to enhance the response to the N stimuli more than to the P stimuli. How might the results be affected if the weights in the PN condition are not fixed to 0.5, and if attention is allowed to differently affect the two types of stimuli? A similar argument could also be made for the weighted sum model.

As you correctly specified, the weighted average model we used is a strict average model, weighted by the attentional bias. We have now added the results for other weighted average variants with different weights for the two stimuli in the absence of attention, and with different attention factors for the preferred and null stimuli.

First, to explore how unequal weights affect the fit of the weighted average model, we tested a weighted average variant with unequal weights (weighted average UW model). Comparison of the fit of this model with the weighted average model with equal weights (weighted average EW) showed that the UW variant was a significantly better fit than the EW model in all regions (*ts* > 3.9, *ps* < 4.7⨯10^-3^, corrected) except in LO, where it was a marginally better fit (*t*(18) = 2.83, *p* = 0.054, corrected).

In the next step, to examine the effect of unequal weights and attention parameters on the fit of the weighted sum and the weighted average models, we tested a weighted average model variant with unequal weights and unequal β parameters for the P and N stimuli (weighted average UWUB). In this variant, no constraint was put on the sum of the weights. Thus, this model is effectively a generalization of the weighted sum and the weighted average models with four parameters. This model was a better fit than the weighted average EW model in all regions (*ts* > 3.78, *ps* < 0.007, corrected) except in EBA (*t*(18) = 2.65, *p* = 0.08, corrected).

We also ran a 4⨯5 ANOVA to compare all weighted average variants with the normalization model. There was a significant effect of model (*F*(3,54)=89.75 , *p*=6.05⨯10^-16^), a significant effect of ROI (*F*(4,72)=34.97 , *p*=2.29⨯10^-10^), and a significant model by ROI interaction (*F*(12,216)=7.55 , *p*=2.5⨯10^-5^). Post-hoc t-tests showed that these weighted average variants were still significantly worse fits to the data than the normalization model in all regions (ps < 7.84⨯10^-4^, corrected) except for EBA, where the normalization model was marginally better than the weighted average UWUB ( *p*=0.065, corrected).

We have included the results of these model comparisons in the *Results section* in lines 320-344, and in figure 5a (see the figure in the response to comment 11 of *Theoretical framework and interpretation* section of this letter):

“The weighted average model we used in previous sections had equal weights for the preferred and null stimuli, with attention biasing the attended preferred or null stimulus with the same amount (the weighted average EW model). However, different stimuli might have different weights in the paired response depending on the neurons' preference towards the stimuli. Besides, attention may bias preferred and null stimuli differently. Therefore, to examine the effect of unequal weights and attention parameters on the fits of the weighted average model, we tested two additional variants of this model.

To examine how unequal weights affect the fit of the weighted average model, we tested the weighted average UW model. Comparison of the fit of this model with the weighted average EW model showed that the UW variant was a significantly better fit than the EW model in all regions (ts > 3.9, ps < 4.7⨯10^-3^, corrected) except in LO, where it was a marginally better fit (t(18) = 2.83, p = 0.054, corrected).

In the next step, to examine the effect of unequal weights and attention parameters on the fit of the weighted sum and the weighted average models, we tested the weighted average UWUB variant. This model had unequal weights and unequal attention parameters for the P and N stimuli. In this variant, no constraint was put on the sum of the weights. Thus, this model was effectively a generalization of the weighted sum and the weighted average models with four parameters. This model was a better fit than the weighted average EW model in all regions (ts > 3.78, ps < 0.007, corrected) except in EBA (t(18) = 2.65, p = 0.08, corrected).

We next compared the goodness of fit of all weighted average variants with the normalization model using a 4⨯5 ANOVA, as illustrated in figure 5a. There was a significant effect of model (F(3,54)=89.75, p=6.05⨯10^-16^), a significant effect of ROI (F(4,72)=34.97, p=2.29⨯10^-10^), and a significant model by ROI interaction (F(12,216)=7.55, p=2.5⨯10^-5^). Post-hoc t-tests showed that these weighted average variants were still significantly worse fits to the data than the normalization model in all regions (ps < 7.84⨯10^-4^, corrected) except for EBA, where the normalization model was marginally better than the weighted average UWUB (p=0.065, corrected).”

We have provided the details of these weighted average variants in the *Methods* section.

(5) Line 57-61: What is the justification of using the weighted sum model when we know that this model is not reflective of what is going on in the brain? It sounds like it has been set up to fail. Has the weighted sum model been shown to predict responses in prior work? It is never really brought up again in the discussion, so its purpose is unclear. Wouldn't a comparison of the weighted average and normalization model make more sense? Better yet, consider comparisons with other model(s) beyond normalization that could explain the data.

Weighted sum and weighted average are two examples of linear summation. It has been previously suggested (by Leila Reddy, 2009) that the response to multiple stimuli lies somewhere between the predictions of these two models, closer to the weighted average. Moreover, Heuer and Britten (2002) and Rubin et al. (2015) have shown that for weak stimuli and in low contrasts, the response to multiple stimuli approaches linearity and can even become supralinear. Since our stimuli were not in full contrast, we had to check for the weighted sum model as well.

In fact, based on our results, the weighted sum model performs better than the weighted average model in V1, and not significantly different from the weighted average model in pFs and PPA. Therefore, for the stimuli we used, the weighted sum model is comparable to the weighted average model in some cases.

We have added this point in the *Results section* in lines 199-206:

“Although many studies have demonstrated that responses to multiple stimuli are added sublinearly in the visual cortex (Heeger 1992, Reddy et al. 2009, Bloem and Ling 2019, Aqil et al. 2021), it has been suggested that for weak stimuli, response summation can approach a linear or even a supralinear regime (Heuer and Britten 2002, Rubin et al. 2015).

Since the stimuli we used in this experiment were presented in a semi-transparent form and were therefore not in full contrast, we found it probable that the response might be closer to a linear summation regime in some cases. We therefore used the weighted sum model to examine whether the response approaches linear summation in any region.”

We agree that we need to discuss weighted sum in the discussion as well. We have included why it was necessary to check the weighted sum model in the *Discussion section* in lines 416-430:

“Stimulus contrast has also been shown to have a crucial role in how single-stimulus responses are added to obtain the multiple-stimulus response. While responses to strong high-contrast stimuli are added sublinearly to yield the multiple-stimulus response, as predicted by the normalization model and the weighted average model, the sublinearity decreases for lower contrasts and even changes to linearity and supralinearity for weak stimuli (Heuer and Britten 2002, Rubin et al. 2015). Here, since the stimuli we used were not in full contrast, we tested the weighted sum model as well to examine whether responses approach linearity in any region. Our results demonstrate that while the weighted average model generally performs better than the weighted sum model in the higher-level occipito-temporal cortex, the weighted sum model provides better predictions in V1. These results suggest stronger sublinearity in higher regions of the visual cortex compared to V1, which is in agreement with previous reports (Kay et al. 2013b). This observation might be related to the higher sensitivity of V1 neurons to contrast (Goodyear and Menon, 1998), causing a more significant decrease in V1 responses to low-contrast stimuli. This, in turn, might make the low-contrast stimulus weaker for V1 neurons, causing a move towards a lower level of sublinearity (Sceniak et al. 1999).”

Regarding comparisons with other models beyond normalization, we have included a linear model with no constraint on the sum of weights, as the generalization of the weighted sum and the weighted average models, with its nonlinear variant with an extra saturation parameter (as explained in response to the previous point). We have included the results of all these models in the *Results section*, and in figure 5 (see the figure in the response to comment 11 of *Theoretical framework and interpretation* section of this letter).

Methods and data analysis(1) FIGURE 1 – protocol – Is task difficulty equated across house category, body category, and color change at fixation?

As shown in figure 1—figure supplement 1 now, task difficulty (reaction time and accuracy) for single-stimulus conditions was equal across house, body, and fixation color blocks. For paired-stimulus conditions, accuracy was the same for B^at^H and BH^at^ conditions, but accuracy was higher in the fixation color change block compared to the BH^at^ condition. However, since accuracies were very high for all participants and across all blocks, this difference is unlikely to have affected our results.

The following information has been added to the *Results section* in lines111-117:

“Overall, average accuracy was higher than 86% in all conditions. Averaged across participants, accuracy was 94%, 89%, 86%, 93%, 94%, 96%, 95% and 96% for B^at^, B^at^H, BH^at^, H^at^, B, H, and BH conditions and the fixation block with no stimulus, respectively. A one-way ANOVA test across conditions showed a significant effect of condition on accuracy (F(7,126) = 8.24, p = 1.63⨯10^-5^) and reaction time (F(7,126) = 22.57, p = 4.52⨯10^-12^). As expected, post-hoc t-tests showed that this was due to lower performance in the B^at^H and BH^at^ conditions (see figure 1—figure supplement 1). There was no significant difference in performance between all other conditions (ps > 0.07, corrected).”

(2) Did participants complete any practice trials for the one back task, or just went right into the scanner? Was the accuracy of the participants in the one back task recorded to ensure that they were paying attention and completing the task properly? Same for the fixation task

Yes, all participants completed one practice run before data collection. If the average accuracy was lower than 75%, they completed another practice run. We only proceeded with data collection if their performance improved to a level higher than 75%, otherwise, no data was recorded. All participants passed this test. As mentioned in response to the previous comment, the actual accuracies inside the scanner were above 86%. We have now reported average accuracy in the *Results section*. We have also included a figure illustrating the accuracy and reaction time for each condition in figure 1—figure supplement 1.

(3) Most of page 6 is a repetition of the methods (much copy and paste), not really needed, instead a short summary would be better or just move the methods to after the introduction.

We have summarized this section in the *Results section*, removing all equations except for one for each model as an example to reduce redundancy.

(4) Some of the details regarding the actual design of the fMRI experiment seem glossed over. From what I can gather, each block was only a few seconds, with no rest between blocks? If so, the BOLD response would be exceptionally driven to saturating levels throughout the experiment. That said, it is difficult to unpack the details based on the Methods section as vague detail is provided.

We apologize if the details were not clear enough. Each block started with a 1-s cue, 1-s fixation, and 8 seconds of stimulus presentation. There was an 8-s fixation period between blocks. In addition, each run started with an 8-s fixation, and there was a final 8-s fixation period after the presentation of the last block in the run. We have added the missing information about the design of the experiment and have modified the design description so that the experiment details are clear to the reader.

(5) Lines 47-51. It is unclear how preferred (P) or null (N) stimulus categories were assigned to each voxel. Does a voxel preferring houses respond greater to houses than a body? Was there a statistical measure for determining what was the P or N stimulus for a given voxel? We imagine that almost all voxels in EBA significantly preferred bodies, and almost all voxels in PPA preferred houses. If so, how is this accounted for? Was there a reliable, statistically significant difference in house or body preference for voxels in the other ROIs, or is the difference between response activation to the two categories marginal? How many voxels preferred houses vs. bodies in each ROI? For each voxel, how reliable is the attribution of the P or N stimulus (e.g., did you use a cross-validation approach and determine P/N on some runs and tested the reliability on the remaining runs)?

We assigned P and N categories to voxels based on their response in isolated house and body conditions. Therefore, for a voxel with higher response to an isolated house compared to an isolated body, house was determined as the preferred category and body as the null category. No statistical measure was used in assigning preferred and null categories to voxels. We have added a new figure in the paper, illustrating the percentage of voxels in each ROI that preferred houses and bodies (figure 2f). As you have pointed out, it is shown in the figure that most voxels in EBA and PPA preferred bodies and houses, respectively, over the other category. It is noteworthy that not all voxels in EBA are body-selective, nor are all voxels in PPA house-selective, which is due to the variability in fMRI data.

In other ROIs, the average percentages of voxel preference for bodies and houses were closer to each other. As you have suggested, we also checked voxel preference consistency in each ROI across the two halves of the data. The results demonstrate high preference consistency in all regions. We have added the result to figure 2-supplementary file 1.

(6) Were the P and N categories determined on all the data (a risk of double-dipping) or only on one half of the dataset?

P and N categories for each voxel were determined in one half of the data, and the model predictions and r-squared calculations were performed on the other half of the data. In the figures, this procedure has been done twice, once for each half of the data, and the results for the two halves were then averaged and illustrated.

(7) Provide more details of the GLM. What were the regressors?

The information is now added in the *Methods* section in lines 593-598:

“We performed a general linear model (GLM) analysis for each participant to estimate voxel-wise regression coefficients for each of the 8 task conditions. The onset and duration of each block were convolved with a hemodynamic response function and were entered to the GLM as regressors. Movement parameters and linear and quadratic nuisance regressors were also included in the GLM. We then used these obtained coefficients to compare the BOLD response in different conditions in each ROI.”

(8) Lines 79-81: Were the model parameters determined in the second half and predictions made in the first half as well? Would averaging the two directions provide more robust results?

We thank the reviewer for this suggestion. We have now changed the analysis to include results from both halves. The results are very similar. We have changed the figures to report the average of the results for the two halves.

We have added the information in the *Results section* in lines 207-211:

“To compare the three models in their ability to predict the data, we split the fMRI data into two halves (odd and even runs) and estimated the model parameters separately for each voxel of each participant twice: once using the first half of the data, and a second time using the second half of the data. All comparisons of data with model predictions were made using the left-out half of the data in each case. All model results illustrate the average of these two cross-validated predictions.”

We have also added the information in the *Methods* section in lines 662-666:

“We repeated this procedure twice: once using the odd half of the data for parameter estimation and the even half for comparing model predictions with the data, and a second time using the even half of the data for parameter estimation and the odd half for comparison with the model predictions. All figures, including model results, illustrate the average of the two repetitions.”

(9) Category stimuli were fit within a 10.2 degree square of visual angle. The methods describe the V1 localizer as a wedge 60 degree in angle. What was the stimulus extent (i.e. eccentricity) of the wedge? Was it also 10.2 degrees so that the V1 ROI encompassed the same amount of visual space as the category stimuli? Same question for the Category localizer. Were these stimuli also 10.2 degrees? Were the ROIs (at least for V1?) defined within the same eccentricity range so that the ROI analysis was confined to where the category stimuli were presented in the visual field?

We thank the reviewer very much for their comment about the size of the stimuli in the main experiment compared to the localizer tasks. The stimuli presented in the V1 localizer and the category localizer were presented with a diameter of 27.1 and 14.3 degrees of visual angle, respectively. We have added the information in the *Methods* section.

Since the localizer stimuli were larger than the stimuli used in the main experiment, we added a step to our analysis in which we selected the voxels that were significantly active during stimulus presentation compared to the fixation block (with p<0.01) in each ROI to compensate for the difference in the size of the localizer stimuli and the stimuli presented in the main experiment. This voxel selection led to a slight change in the results, most significantly in V1. The normalization model’s fit improved in V1 after this voxel selection. This procedure did not qualitatively change the results in other regions. We have added the details of this extra step to the methods section.

(10) What software was used to define LOC PF EBA and PPA? And what metric do we have to ensure that these are reliable ROIs? The authors would benefit by including figures that show examples of the ROIs overlaid on contrast data for the localizer conditions. The same goes for the β weights within the ROIs.

We used Freesurfer’s tksurfer module. We defined the ROIs for each participant individually based on the voxels that were significantly active in – the contrasts mentioned in the Methods section. The activation maps were thresholded at p<0.001. We have provided examples of the defined ROIs overlaid on contrast data for one of the participants. Since these are standard localizers, and we selected all voxels that passed the threshold as stated in the paper, we do not think adding all individual participants localizer data in the paper would be useful. We have therefore decided not to include them. If the reviewer feels strongly about this, we are happy to add them as a figure supplement to figure 1, See Author response image 1.

**Author response image 1. sa2fig1:** 

(11) Preprocessing – how were the data aligned?

We have added the information in the *Methods* section in lines 574-577:

“The data in each run were motion-corrected per-run and aligned to the anatomical data using the middle time point of that run. The fMRI data from the localizer was smoothed using a 5-mm FWHM Gaussian kernel, but no spatial smoothing was performed on the data from the main experiment to optimize the voxel-wise analyses.”

(12) The paper refers to a Supplementary Methods, which I could not find.

Thank you for catching this typo. We have corrected the sentence in line 217.

Results and statistics(1) What was accuracy like on these tasks? There was no description of performance at all. It is necessary to provide this information to assess the attentional manipulation

We have included the following paragraph in the *Results section* in lines 111-117:

“Overall, average accuracy was higher than 86% in all conditions. Averaged across participants, accuracy was 94%, 89%, 86%, 93%, 94%, 96%, 95% and 96% for B^at^, B^at^H, BH^at^, H^at^, B, H, and BH conditions and the fixation block with no stimulus, respectively. A one-way ANOVA test across conditions showed a significant effect of condition on accuracy (F(7,126) = 8.24, p = 1.63⨯10^-5^) and reaction time (F(7,126) = 22.57, p = 4.52⨯10^-12^). As expected, post-hoc t-tests showed that this was due to lower performance in the B^at^H and BH^at^ conditions (see figure 1—figure supplement 1). There was no significant difference in performance between all other conditions (ps > 0.07, corrected).”

We have also illustrated the accuracies and reaction times of all task conditions in figure 1—figure supplement 1.

(2) Line 96 – What is ts? T-statistic? Did you run a series of t-tests comparing normalization vs noise ceiling? Why not an ANOVA and correct for multiple comparisons? And what does it mean that the R2 was not significant different from noise ceiling? Interpretation of these results must be provided.

We thank the reviewer for pointing out the problem in our analysis. We have now reported the results of ANOVA and post-hoc t-tests with multiple comparison corrections in the *Results section* in lines 229-234:

“We first compared the goodness of fit of the three models across the five ROIs using a 3⨯5 repeated measures ANOVA. The results showed a significant main effect of model (F(2,36) = 72.9, p = 9.86⨯10^-11^) and ROI (F(4,72) = 26.66, p = 1.04⨯10^-7^), and a significant model by ROI interaction (F(8,144) = 24.96, p = 3.74⨯10^-15^). On closer inspection, the normalization model was a better fit to the data than both the weighted sum (ps < 0.019, corrected) and the weighted average (ps < 5.7⨯10^-5^, corrected) models in all ROIs.”

To answer the reviewer’s question about the noise ceiling, we defined the noise ceiling in each region separately as the r-squared of the correlation between the odd and even halves of the data. Given that the correlation between the model and the data cannot exceed the reliability of the data (as calculated by the correlation between the data from odd and even runs), the r-squared can also not exceed the squared split-half reliability. The noise ceiling (squared split-half reliability), therefore, determines the highest possible goodness of fit a model can reach.

We calculated the normalization model’s r-squared difference from the noise ceiling (NRD) for each ROI. NRD is a measure of the ability of the model in accounting for the explainable variation in the data; the lower the difference between the noise ceiling and a model’s goodness of fit, the more successful that model is in predicting the data. We ran a one-way ANOVA to test for the effect of ROI on NRD, and observed that this measure was not significantly different across ROIs (*F*(4,72)=0.58, *p* = 0.61), demonstrating that the normalization model was equally successful across ROIs.

We have included the interpretation of the noise ceiling and its relationship with the models’ goodness of fit in the *Results section* in lines 223-228:

“We also calculated the noise ceiling in each region separately as the r-squared of the correlation between the odd and even halves of the data. Given that the correlation between the model and the data cannot exceed the reliability of the data (as calculated by the correlation between the data from odd and even runs), the r-squared can also not exceed the squared split-half reliability. The noise ceiling (squared split-half reliability), therefore, determines the highest possible goodness of fit a model can reach.”

We have also included the result of the ANOVA test for the difference between the noise ceiling and the normalization model’s r-squared across ROIs in the *Results section* in lines 240-246:

“We then calculated the normalization model’s r-squared difference from the noise ceiling (NRD) for each ROI. NRD is a measure of the ability of the model in accounting for the explainable variation in the data; the lower the difference between the noise ceiling and a model’s goodness of fit, the more successful that model is in predicting the data. We ran a one-way ANOVA to test for the effect of ROI on NRD, and observed that this measure was not significantly different across ROIs (F(4,72)=0.58, p = 0.61), demonstrating that the normalization model was equally successful across ROIs in predicting the explainable variation in the data.”

(3) Lines 119-123. Again, why a t-test statistic here? Did you run multiple t-tests for each ROI? Shouldn't this be an ANOVA across the 5 ROIs? As it stands, line 120 says that you ran multiple t tests across all the regions, and then report single t-statistic? The paper needs to be clear (and correct) in the type of statistics it uses to make its claims.

We have now performed a repeated-measures 2-way ANOVA across the 5 ROIs to compare the predictions of the effects of attention by the three models, corrected for multiple comparisons in all our analyses. Results remain qualitatively similar.

We have included the results in the *Results section* in lines 282-291:

“To compare how closely the predictions of the three models followed the response change in the data, we calculated the difference between the response change observed in the data and the response change predicted by each model. Then, we ran a 3⨯5 repeated measures ANOVA with within-subject factors of model and ROI on the obtained difference values. The results demonstrated a significant effect of model(F(2,36) = 105.59, p = 4.98⨯10^-9^), a significant effect of ROI(F(4,72) = 13.88, p = 4.62⨯10^-6^), and a significant model by ROI interaction (F(8,144) = 28.63, p = 2.13⨯10^-8^). Post-hoc t-tests showed that the predictions of the normalization model were closer to the response change observed in the data in all ROIs ( ps < 5.7⨯10^-7^, corrected ) except in V1, where the predictions of the weighted sum and the weighted average models were closer to the data ( ps < 8.6⨯10^-8^, corrected ).”

And in lines 310-318:

“We calculated the difference between the asymmetry index observed in the data and the predicted index by each model and performed a 3⨯5 repeated measures ANOVA to compare the three models in how closely they predicted the asymmetry effect across ROIs using these difference values. We observed a significant effect of model (F(2,36)=185.3, p = 1.51⨯10^-11^), a significant effect of ROI (F(4,72) = 64.97, p = 3.71⨯10^-15^), and a significant model by ROI interaction (F(8,144) = 45.60, p = 8.97⨯10^-17^). The prediction of the normalization model was closer to the data in all regions ( ps < 1.9⨯10^-7^, corrected ) except for PPA, where the prediction of the weighted sum model was closer to the asymmetry observed in the data than the prediction of the normalization model ( p = 4.37⨯10^-5^, corrected ).”

(4) There is no assessment of the quality of the data – no Β weight maps showing the responses on the brain, or what proportion of voxels in each ROI actually responded to the stimuli. A lot of interesting data that could be shown is reduced to line plots -this comes across as the paper starting at the end, rather than providing any sense of buildup in the data analysis and the conclusions that the authors draw.

We thank the reviewer for this suggestion. We have added a new figure including the average voxel responses in each condition for each ROI and the percentage of voxels in each region that are more responsive to houses and bodies (panel f in figure 2, also pasted in response to comment 9 of the *Theoretical framework and interpretation* section of this letter). We think the addition of this figure helped with the clarification of the later analyses and flow of the manuscript.

(5) The authors show the goodness of fit across the 5 modeled conditions in 2C – fine, but what is the justification for lumping all the conditions together? Is there a supplement figure that teases this apart, with the goodness of fit for each model for each of the conditions, for each ROI? It would help to be comprehensive.

We apologize if the method was not clear. We defined the goodness of fit across conditions for each voxel separately (and not across voxels within each condition). Therefore it is not possible to show the goodness of fit for each condition. The reason for calculating the goodness of fit in this manner was to evaluate model fits based on their ability to predict response changes with the addition of a second stimulus and with the shifts of attention. Since correlation is blind to a systematic error in prediction for all voxels in a condition, calculating the goodness of fit across voxels would lead to misinterpretation. See figure 3—figure supplement 1 for an example figure showing how we got the goodness of fit in one voxel across all conditions.

The details of r-squared calculation were provided before in the *Results* and *Methods* sections. We have now included a reference to this figure in the *Results section* in lines 220-222:

“We calculated the goodness of fit for each voxel by taking the square of the correlation coefficient between the predicted model response and the respective fMRI responses across the five modeled conditions (figure 3—figure supplement 1).”

and in the *Methods* section in lines 669-671:

“The goodness of fit was calculated by taking the square of the correlation coefficient between the observed and predicted responses for each voxel across the five modeled conditions (figure 3—figure supplement 1).”

(6) Averaging all the data together (e.g., in Figure 2c across all conditions) does not provide a clear picture of the results. Is it possible that it is only in the attention conditions that the normalization model outperforms the others (although again, note the remark about the different numbers of free parameters in each model)? It would be more transparent to plot Figure 2c separately for the isolated/paired conditions, even though the story might become more complex.

As we pointed out in our response to the previous comment, the goodness of fit was calculated across the five conditions (as illustrated in (figure 3—figure supplement 1)). Since there is only one condition with unattended stimuli, it is not possible to calculate the goodness of fit across one condition only. Please note that we have provided the prediction of each condition separately in figure 3a-e. However, we have compared the results of the PN condition with model predictions and have added the statistics of this comparison in the *Results section* in lines 247-257:

“Interestingly, just focusing on the paired condition in which none of the stimuli were attended (the PN condition), the results of the weighted average model were closer to the normalization model (the gray and orange isolated data points on the subplots a-e of figure3 are similarly close to the navy point of data in some regions). For this condition, the predictions of the normalization model were significantly closer to the data compared to the predictions of the weighted average model in V1, pFs, and PPA (ps < 0.028, corrected) but not significantly closer to the data in LO and EBA (ps > 0.09, corrected). These results are in agreement with previous studies suggesting that the weighted average model provides good predictions of neural and voxel responses in the absence of attention (Zoccolan et al. 2005, Macevoy and Epstein 2009, Kliger and Yovel 2020). However, when considering all the attended and unattended conditions, our results show that the normalization model is a generally better fit across all ROIs.”

Regarding your point about the different number of parameters in the models, as explained in points 1, 2, and 4 of the *Model* section of this letter, we have used new models with more free parameters, AIC calculation, and simulations to show that the difference in the number of parameters cannot account for the observed success of the normalization model compared to other models.

(7) Related to the above points, the data are provided in a very synthesized manner, and raw results are not shown. For example, could we see the values of the estimated model parameters? How do they compare when determined in each half of the data (to get a sense of reliability)?

We have included the values of the estimated parameters for the even and odd runs for each model in supplementary files 3-8. As shown in the tables, the values of the estimated parameters in the two halves are very close to each other.

We have also added a figure including the average fMRI regression coefficients in each condition (figure 2), voxel preference consistency across odd and even runs (figure 2—figure supplement 1), and fMRI regression coefficients plotted separately for odd and even runs (figure 2—figure supplement 1) to provide the readers with a sense of reliability and data quality.

Figures(1) Figure 1b: The gray squares overlap a lot so that we don't see the stimuli in each gray square.

We have reduced the overlap between the gray squares, so the stimuli are fully shown.

(2) The y-axis label in Figure 2A is not BOLD response (this would be appropriate if the figure was a BOLD time series); is this the average fMRI Β weight?

Yes, we used the fMRI Β weights. But in order to not confuse the GLM Β weights with the attention-related Β values in the models and to satisfy the reviewer’s concern, we opted to use “fMRI regression coefficient” in this plot.

(3) In Figure 2 (a and b), the connecting of data points as if they are on a continuum is not correct given their categorical nature. Moreover, the figure could be improved so that the actual data points are made more prominent, and the model predictions are dashed lines. What exactly do the error bars in Figure 2a and 2b correspond to?

Although responses in the seven conditions are discrete and not continuous, we have connected the responses in attended conditions (in which body or house stimuli were attended) and unattended conditions (in which body and house were ignored and the fixation point color was attended) separately. This was done, first, to avoid the confusion caused by illustrating too many single points in the figure and, second, to better visualize the effects of attention. As you can see in Author response image 2, removing the lines will render the plot incomprehensible.

We have added the justification for connecting data points in the *Results section* in lines 155-159:“Note that although the seven conditions constitute a discrete and not a continuous variable, we have connected the responses in attended conditions (in which body or house stimuli were attended) and unattended conditions (in which body and house were ignored and the fixation point color was attended) separately. This was done for visual purposes and ease of understanding. ”

Regarding your suggestion on making the data points more prominent, we have thickened the lines related to the actual data.

Error bars represent standard errors of the mean for each condition, calculated across participants after removing the overall between-subject variance. We have added the information in the figure caption.

(4) Suggestion to order the panels in Figure 2 logically (the BOLD and model data all presented in A for all 5 ROIs).

We have now ordered the panels logically according to the reviewer’s suggestion.

(5) Is there any statistic to help evaluate the data in Figure 2C? (Line 93)

We have now added the statistics in the *Results section* in lines 229-234:

“We first compared the goodness of fit of the three models across the five ROIs using a 3⨯5 repeated measures ANOVA. The results showed a significant main effect of model (F(2,36) = 72.9, p = 9.86⨯10^-11^) and ROI (F(4,72) = 26.66, p = 1.04⨯10^-7^), and a significant model by ROI interaction (F(8,144) = 24.96, p = 3.74⨯10^-15^). On closer inspection, the normalization model was a better fit to the data than both the weighted sum (ps < 0.019, corrected) and the weighted average (ps < 5.7⨯10^-5^, corrected) models in all ROIs.”

(6) FIGURE 3 – Why not include the BOLD response for P only and N only? That would be informative.

The BOLD response for isolated P and isolated N conditions is shown in figure 2a-e (now changed to figure 3a-e because we have added a new figure).

The purpose of figure 3 (now figure 4) is to explain the two features of attention, which is related to the conditions with attention directed to one of the stimuli. We agree that including the P and N conditions along with attended conditions would be informative, and this is already provided in figure 2 (now figure 3) for all regions. Figure 3 brings only an example to better visualize how each index is calculated.